EMBO
Molecular Medicine

# PARP-1 regulates DNA repair factor availability

Matthew J Schiewer[1,2,*], Amy C Mandigo[1,2], Nicolas Gordon[1,2], Fangjin Huang[3], Sanchaika Gaur[3], Renée de Leeuw[1,2], Shuang G Zhao[4], Joseph Evans[4], Sumin Han[4], Theodore Parsons[2,5], Ruth Birbe[6], Peter McCue[2,5], Christopher McNair[1,2], Saswati N Chand[1,2], Ylenia Cendon-Florez[1,2], Peter Gallagher[1,2], Jennifer J McCann[1,2], Neermala Poudel Neupane[1,2], Ayesha A Shafi[1,2], Emanuela Dylgjeri[1,2], Lucas J Brand[1,2], Tapio Visakorpi[7], Ganesh V Raj[8], Costas D Lallas[2,9], Edouard J Trabulsi[2,9], Leonard G Gomella[2,9], Adam P Dicker[2,10], Wm. Kevin Kelly[2,11], Benjamin E Leiby[2,12], Beatrice Knudsen[3] (iD), Felix Y Feng[13] & Karen E Knudsen[1,2,9,10,11]

## Abstract

PARP-1 holds major functions on chromatin, DNA damage repair and transcriptional regulation, both of which are relevant in the context of cancer. Here, unbiased transcriptional profiling revealed the downstream transcriptional profile of PARP-1 enzymatic activity. Further investigation of the PARP-1-regulated transcriptome and secondary strategies for assessing PARP-1 activity in patient tissues revealed that PARP-1 activity was unexpectedly enriched as a function of disease progression and was associated with poor outcome independent of DNA double-strand breaks, suggesting that enhanced PARP-1 activity may promote aggressive phenotypes. Mechanistic investigation revealed that active PARP-1 served to enhance E2F1 transcription factor activity, and specifically promoted E2F1-mediated induction of DNA repair factors involved in homologous recombination (HR). Conversely, PARP-1 inhibition reduced HR factor availability and thus acted to induce or enhance "BRCA-ness". These observations bring new understanding of PARP-1 function in cancer and have significant ramifications on predicting PARP-1 inhibitor function in the clinical setting.

**Keywords** DNA repair; E2F1; PARP; transcription
**Subject Category** Cancer

## Introduction

Poly(ADP-ribose) polymerase 1 (PARP-1) is a multifunctional protein of importance in cancer. PARP-1 is an abundantly expressed nuclear enzyme which uses $NAD^+$ as a substrate to poly(ADP-ribose)-ylate (PARylate) nuclear proteins, including automodification of PARP-1 itself (D'Amours *et al*, 1999; Krishnakumar & Kraus, 2010). PARP-1 plays a key role in several key biological processes: replication fork stability (Bryant *et al*, 2009), cell death (Yu *et al*, 2002), DNA repair and genomic stability (Durkacz *et al*, 1980), telomere maintenance (Beneke *et al*, 2008), chromatin organization (Poirier *et al*, 1982), and transcriptional regulation (Kraus & Lis, 2003; Schiewer & Knudsen, 2014).

The DNA repair functions of PARP-1 have been targeted for anti-cancer effects through use of pharmacological PARP inhibitors (PARPi; Lord & Ashworth, 2008), which have been approved of ovarian cancer, and are under clinical investigation in a number of other tumor types, including prostate cancer (PCa). It is thought that PARPi are especially effective in tumors that lack homologous recombination (HR) capacity through loss-of-function mutations in *BRCA1* or *BRCA2*, in a phenomenon termed synthetic lethality (McCabe *et al*, 2006; Lord & Ashworth, 2017). However, clinical trial data in *BRCA1/2* mutant-selected tumors indicate that objective response rates are only ~ 40%, suggesting that *BRCA1/2* mutation is not sufficient for PARPi response (Fong *et al*, 2009; Audeh *et al*, 2010; Gelmon *et al*, 2011; Kaye *et al*, 2012; Sandhu *et al*, 2013; Coleman *et al*, 2015). Additionally, a recently published clinical trial

1  Department of Cancer Biology, Thomas Jefferson University, Philadelphia, PA, USA
2  Sidney Kimmel Cancer Center, Thomas Jefferson University, Philadelphia, PA, USA
3  Cedars-Sinai Medical Center, Los Angeles, CA, USA
4  Department of Radiation Oncology, University of Michigan, Ann Arbor, MI, USA
5  Department of Pathology, Thomas Jefferson University, Philadelphia, PA, USA
6  Cooper University Health, Camden, NJ, USA
7  University of Tampere, Tampere, Finland
8  UT Southwestern, Dallas, TX, USA
9  Department of Urology, Thomas Jefferson University, Philadelphia, PA, USA
10 Department of Radiation Oncology, Thomas Jefferson University, Philadelphia, PA, USA
11 Department of Medical Oncology, Thomas Jefferson University, Philadelphia, PA, USA
12 Department of Pharmacology and Experimental Therapeutics, Thomas Jefferson University, Philadelphia, PA, USA
13 Departments of Radiation Oncology, Urology, and Medicine, University of California, San Francisco, San Francisco, CA, USA
   *Corresponding author. Tel: +1 215 503 8574; E-mail: karen.knudsen@jefferson.edu

combining PARPi and androgen receptor (AR)-directed therapy in patients with advanced PCa demonstrated clinical benefit, irrespective of HR status (Clarke *et al*, 2018). Furthermore, the TO-PARP trial (Mateo *et al*, 2015) led to FDA Breakthrough Status for patients with *BRCA2* or *ATM* mutant castration-resistant prostate cancer (CRPC). Olaparib responders were enriched for defects in DNA repair genes, such as biallelic loss of *BRCA2* and *ATM*. However, while most responders (14/16) in this trial were categorized as biomarker positive for HR deficiency, the biomarker suite included single copy loss of DNA repair factors, as well as alterations to *HDAC2*, which is involved in transcriptional repression (Rountree *et al*, 2000). While these studies that not all PARPi responders with PCa harbor HR-defective tumors, and not all PCa tumors that exhibit aberrant DNA repair are PARPi responsive, there is clinical evidence that PARPi resistance is associated with restored HR function in multiple tumor types (Edwards *et al*, 2008; Barber *et al*, 2013; Christie *et al*, 2017; Kondrashova *et al*, 2017; Pishvaian *et al*, 2017; Weigelt *et al*, 2017), including PCa (Goodall *et al*, 2017; Quigley *et al*, 2017). Additionally, PARPi resistance has been associated with differential DNA damage response (DDR) network functioning (Jaspers *et al*, 2013; Johnson *et al*, 2013; Gogola *et al*, 2018). These mechanisms of resistance to PARPi indicate that for these tumors, DDR defects likely led to PARPi responses. These clinical findings indicate that further mechanistic understanding of PARP-1 functions is needed to develop useful clinical biomarkers of response to PARPi.

Given the potential implications of PARP-1-mediated functions in human malignancies, and the need for biomarkers of PARPi response, it was imperative to discern the molecular basis of PARP-1 function and activity in the context of *BRCA1/2* wild-type PCa, and determine the contribution of PARP-1-mediated transcriptional events on tumor phenotypes.

# Results

## PARP-1 enzymatic activity is increased as a function of disease progression and is associated with poor outcome

To ascertain the impact of PARP-1 function on aggressive tumor behavior, PCa was utilized as a disease system. In this tumor type, the role of PARP-1 in transcriptional regulation of key transcription factors of PCa relevance has been demonstrated (ETS transcription factors and androgen receptor (AR); Brenner *et al*, 2011; Schiewer *et al*, 2012), and AR is a key driver of PCa initiation and progression. Furthermore, PARPi has generated promising clinical trial data in advanced PCa (Mateo *et al*, 2015). Initially, human tissues from primary, hormone therapy (HT)-sensitive PCa, and metastatic CRPC (mCRPC) were queried for PARP-1 enzymatic activity via immunohistochemistry (IHC) for PAR (Poly(ADP-ribose), the product of PARP-1 enzymatic activity; Fig 1A). PARP-1 enzymatic activity was elevated in mCRPC when compared to primary PCa (Fig 1B). These data give confirmation of predictions from preclinical models which showed elevated PARP-1 enzymatic activity in CRPC cell lines (including C4-2 and LNCaP-abl) compared to hormone therapy (HT)-sensitive cell lines (including LNCaP, LAPC4, and VCaP; Schiewer *et al*, 2012). To query the impact of elevated PARP-1 enzymatic activity on clinical outcomes, PARP-1 activity was assessed as a

function of proliferative indices (Appendix Fig S1A) and cT stage at primary diagnosis (Appendix Fig S1B). No correlation was observed, indicating that higher PARP-1 activation status is not simply due to increased cell proliferation or larger volume tumor. Furthermore, there were no correlations between PARP-1 enzymatic activity and molecular alterations that are frequent in PCa, including *TMPRSS2:ERG* fusion status (Appendix Fig S1C), PTEN score (Appendix Fig S1D), or *AR* copy number (Appendix Fig S1E). However, enhanced PARP-1 activity was significantly associated with decreased progression-free survival (PFS; Fig 1C). These data indicate that PARP-1 enzymatic function is not only elevated in CRPC, but also predictive of PFS, which is associated with disease-specific mortality.

To expand upon these data, multiplexed quantifiable immunofluorescent IHC was performed on non-neoplastic prostate tissue, primary PCa, and mCRPC (Fig 1D top left, higher magnification at right). As measured through quantification of PAR immunoreactivity, PARP-1 enzymatic activity was elevated in primary PCa (median value 62.03) as compared to non-neoplastic prostate tissue (median value 51.52), and highest in mCRPC tissue (median value 69.10; Fig 1D, bottom left). However, the observed increase in PARylation during disease progression cannot be simply attributed to total PARP-1 protein expression, as the ratios of PARP-1 and PAR expression levels differed across disease states (Fig 1D, bottom middle; medians of PAR values 51.67, 54.29, and 47.81 for non-neoplastic, primary PCa, and mCRPC, respectively).

Being intricately involved in DNA damage repair, PARP-1 enzymatic activity is induced by DNA damage (Durkacz *et al*, 1980). To determine whether the elevated PARP-1 enzymatic activity in mCRPC observed above was associated with DNA damage repair, immunoreactivity of γH2AX, a measure of repair of DNA double-strand breaks (DSBs; Podhorecka *et al*, 2010), was performed. This analysis indicated that PARP-1 enzymatic activity as a function of disease progression was not associated with repair of DSBs (Fig 1D, bottom right; median values 44.20, 51.80, and 46.20 for non-neoplastic, primary PCa, and mCRPC, respectively), suggesting that PARP-1 activity is regulated by other factors in addition to DNA damage. Dual assessment of DSB repair and PARP-1 activity in each specimen revealed a positive correlation between PAR and γH2AX in non-neoplastic prostate tissues ($r = 0.2853$), and primary PCa tissues ($r = 0.3573$), but this association is lacking in mCRPC tissues ($r = -0.03825$; Fig 1E), further indicating that elevated PARP-1 enzymatic function in mCRPC is not attributable to increased DNA DSB repair. Together, these data demonstrate that PARP-1 enzymatic activity is heterogeneous, increases as a function of PCa progression, is not associated with levels of either PARP-1 protein expression or of DNA damage repair in mCRPC, and may predict poor outcome in PCa.

## Identification of the PARP-1-regulated transcriptome and relevance for disease progression

As demonstrated above, PARP-1 enzymatic activity is elevated as a function of PCa progression independent of DNA DSB repair. As such, other PARP-1 functions were analyzed. To assess PARP-1-mediated transcriptional regulation in the context of androgen signaling, hormone therapy-sensitive (HT-sensitive) PCa cells were deprived of steroids for 72 h, then treated with PARP-1 inhibition

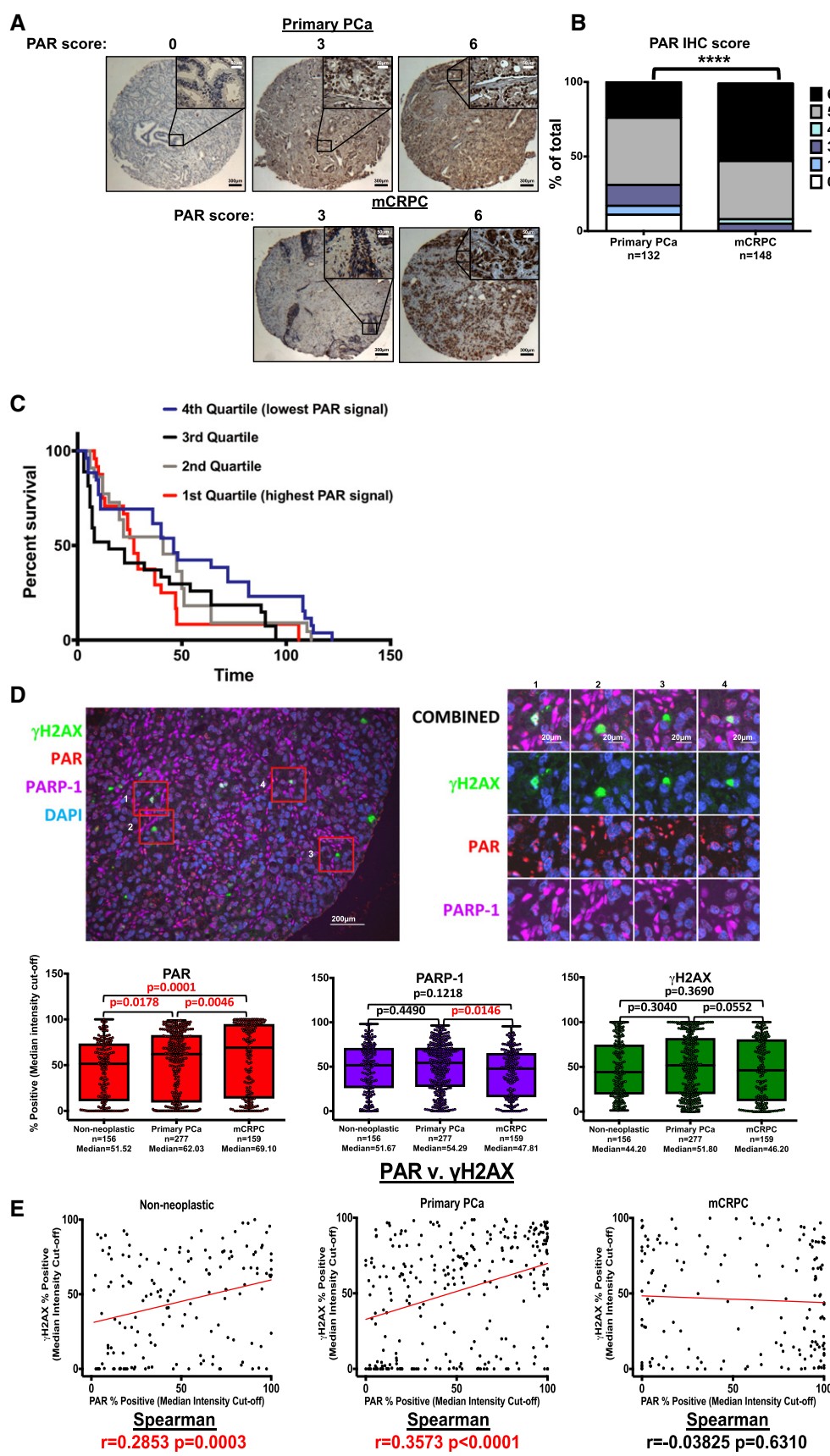

**Figure 1.**

◄

**Figure 1.  PARP-1 enzymatic activity is increased as a function of disease progression and is associated with poor outcome.**

A  Tissue microarrays (TMAs) from primary PCa (n = 132) and CRPC (n = 148) were stained via immunohistochemistry for poly(ADP-ribose; PAR), and scored by a clinical pathologist (T. Parsons) for intensity (0–3) and percentage (0–3).

B  PAR score was generated via the equation: (intensity × 1) + (percentage × 2). PAR scores were compared between primary and CRPC. ****P value < 0.0001 by Chi-square test.

C  Manual PAR scores were divided in to quartiles and then were compared to progression-free survival in the CRPC TMAs. *P < 0.05, ns = not statistically significant by Log-rank (Mantel-Cox). 1st quartile vs. 2nd quartile, P = 0.1482; 1st quartile vs. 3rd quartile, P = 0.5794; 1st quartile vs. 4th quartile, P = 0.0160; 2nd quartile vs. 3rd quartile, P = 0.3869; 2nd quartile vs. 4th quartile, P = 0.2110; 3rd vs. 4th quartile, P = 0.0201. 1st quartile (n = 24); 2nd quartile (n = 22); 3rd quartile (n = 27); 4th quartile (n = 26).

D  Top left: Representative image of one TMA core after multiplex fluorescent IHC for γH2AX (green), PAR (red), PARP-1 (purple), with DNA (blue). Top right: Insets of parent image on the left. Numbers above inset columns coincide with numbers on image at left that were chosen for further magnification and representation (boxed areas). Bottom left: Percent positive staining for PAR for the entirety of each TMA cohort. Bottom middle: Percent positive staining for PARP-1. Bottom right: γH2AX for the entirety of each TMA cohort. Data were considered after a median intensity cutoff and analyzed for statistical significance using two-tailed Student's t-test for PAR, PARP-1, and γH2AX, respectively. Exact P values are indicated. Horizontal lines are median. Box limits are 25% and 75% percentiles, and whiskers are min to max.

E  Two-tailed Spearman correlation test between PAR and γH2AX (% positive with a median intensity cutoff). Exact P values are indicated when available.

Source data are available online for this figure.

(or control) followed by 16 h dihydrotestosterone (DHT) stimulation as depicted in Fig 2A (left) to assess the impact of PARP-1 suppression in the presence and absence of AR activity. As expected, DHT altered the expression of a large number of transcripts (n = 1358), and PARP-1 suppression resulted in differential transcript expression when compared to DHT (n = 877; Fig 2A, right), consistent with previous reports that PARPi alters the transcriptional effects of androgen signaling (Schiewer *et al*, 2012). This was further confirmed using a previously characterized set of AR/DHT-responsive target genes, the majority of these genes are oppositely regulated by DHT and PARPi in LNCaP (Appendix Fig S2A). PARP-1 has also been found to regulate castration-resistant AR function (Schiewer *et al*, 2012). To assess the overall transcriptional effects of PARP-1 in an unbiased manner in the context of CRPC, C4-2 cells were deprived of steroids for 72 h, then were either treated PARPi (or control) as depicted in Fig 2B (left) for 16 h. In total, 2011 transcripts were differentially regulated upon PARPi treatment when compared to control in CRPC cells, thus defining a PARP-1-regulated transcriptome in CRPC. The overlap of differentially regulated genes in HT-sensitive vs. CRPC cells was derived, and the data indicate there are both overlapping and distinct transcriptional changes elicited by each condition and in the individual cell lines (Appendix Fig S2B). Gene lists are included in Dataset EV1. These data indicate that there may be a core transcriptional program regulated by PARP-1 in PCa cells, which includes a large number of DHT-responsive genes (n = 169), but the transition to castration resistance likely expands the relevance of PARP-1-regulated transcription, given the larger number of transcripts that are altered upon PARPi (n = 1,810 unique genes regulated by PARP-1). Importantly, the transcripts associated with active PARP-1 (down-regulated by PARPi) in both HT-sensitive and CRPC cells significantly increased in expression from benign tissues, to primary PCa, to PCa metastases (Fig 2C) when these transcripts were queried against a publically available data set (Grasso *et al*, 2012). Furthermore, these data were validated using other publically available data sets (Lapointe *et al*, 2004; Taylor *et al*, 2010; Yu *et al*, 2007; Appendix Fig S3), thus indicating that the PARP-1-responsive transcriptome is elevated as a function of PCa progression. Together with immunohistochemical PARP-1 activity assessment (Fig 1), these collective data indicate that both PARP-1 enzymatic activity and PARP-1-sensitive transcriptional events are enhanced as a function of disease progression.

**PARP-1 regulates pro-oncogenic transcription factor signaling**

To assess the potential biological consequences of the observed transcriptional enhancement of PARP-1, Gene Set Enrichment Analysis (GSEA; Mootha *et al*, 2003; Subramanian *et al*, 2005) Molecular Signatures Database (MSigDB) analyses were performed using the unbiased data generated as described above. Utilizing the generalizable KEG MSigDB demonstrated an enrichment for cell cycle-related and DNA damage repair-associated pathways (including homologous recombination; Fig 3A, left). Analyses using the more specific Hallmarks MSigDB confirmed previous studies, in that the Androgen Response hallmark was enriched in and suppressed in CRPC cells (NES = −2.54; Fig 3A, right bottom). The statistically highest enriched MSigDB hallmark was E2F Targets (HT-sensitive NES = −1.51, CRPC NES = −3.31; Fig 3A, right top), which has canonical roles in the regulation of both the cell cycle and DNA damage repair (Biswas & Johnson, 2012). These data indicate that in addition to playing a key role in AR transcriptional activity, PARP-1 transcriptionally regulates processes associated with the cell cycle and DNA damage repair.

The E2F family of transcription factors regulate critical processes of importance in cancer, including cell cycle regulation, DNA repair (Biswas & Johnson, 2012), mitochondrial function (Goto *et al*, 2006), cell death (Polager & Ginsberg, 2009), tumor progression and metastatic development (Alla *et al*, 2010), stemness (Chen *et al*, 2008, 2009), and angiogenesis (Qin *et al*, 2006). E2F1 is frequently deregulated in PCa (Sharma *et al*, 2010), and deregulated E2F1 activity is associated with aggressive disease (McNair *et al*, in press, JCI). For validation, both HT-sensitive and CRPC cells were treated as depicted in Fig 2A and B above, RNA was extracted, and subjected to qPCR for canonical E2F1 target genes (*E2F1*, *PCNA*, *MCM7*, and *CCNA2*). As shown, each of these transcripts was diminished by treatment with the PARPi veliparib by 40–60% in both the context of HT-sensitive (Fig 3B, top) and CRPC cells (Fig 3B, bottom). Confirmation that these genes are E2F1 target genes was conducted by transiently knocking down E2F1, and subsequent gene expression analyses (Appendix Fig S4A). To explore the impact of exogenous E2F1 expression on PARP-1-regulated E2F1 activity, models of exogenous E2F1 were generated. Upon examination of E2F1 target gene expression after PARP inhibition (Appendix Fig S4B), it was determined that E2F1 target gene expression is no longer under the control of PARP-1. These data indicate that

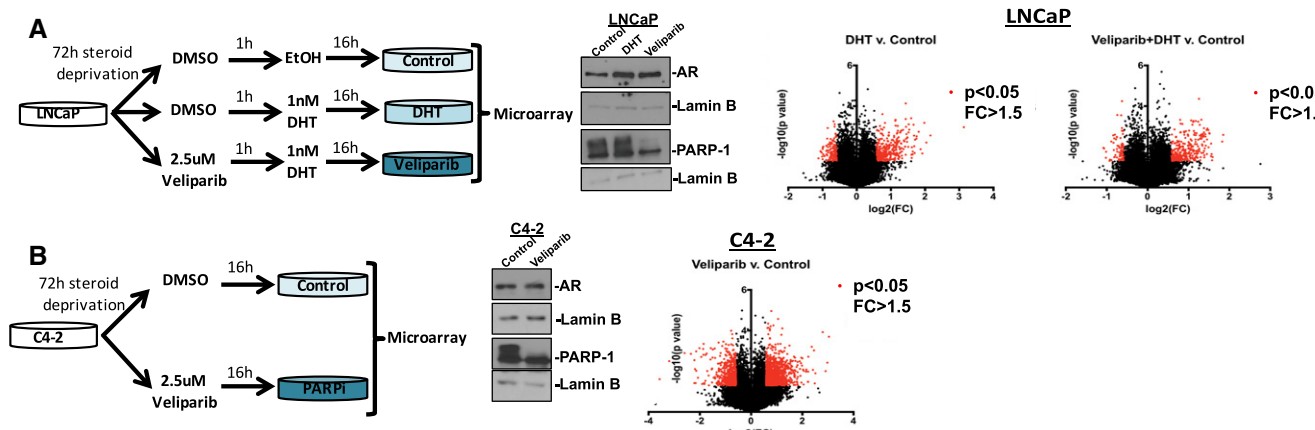

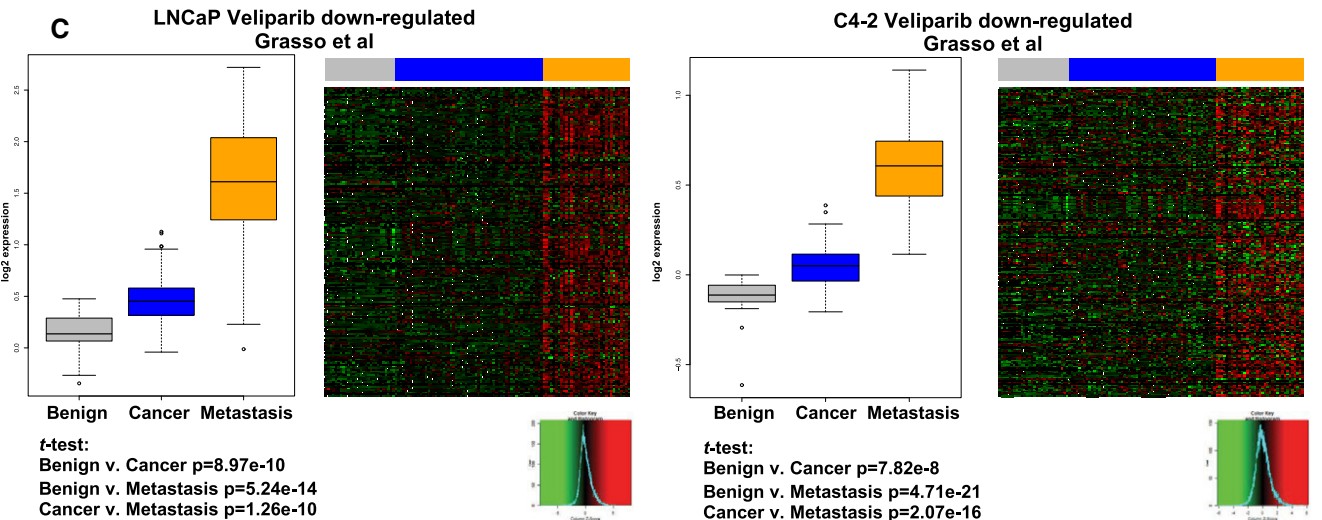

**Figure 2. Identification of the PARP-1-regulated transcriptome and relevance for disease progression.**

A Left: Schematic representing the conditions utilized for transcriptomic analyses (n = 2) of HT-sensitive LNCaP cells. Cells were deprived of hormones for 72 h, followed by either treatment with 2.5 μM veliparib (PARPi) or vehicle control (DMSO) for 1 h, then subsequently treated with either 1 nM DHT or vehicle control (EtOH) for 16 h. Middle: Immunoblot with the indicated antisera. Right: Volcano plots of transcripts found to be differentially regulated by DHT vs. EtOH (left) or DHT vs. PARPi followed by DHT (right). Red dots indicate transcripts that were both statistically significantly altered (P < 0.05) and more than 1.5-fold changed.

B Left: Schematic representing the conditions utilized for transcriptomic analyses (n = 2) of CRPC C4-2 cells. Cells were deprived of hormones for 72 h, followed by either treatment with 2.5 μM veliparib (PARPi) or vehicle control (DMSO) for 16 h. Middle: Immunoblot with the indicated antisera. Right: Volcano plots of transcripts found to be differentially regulated PARPi vs. vehicle control. Red dots indicate transcripts that were both statistically significantly altered (P < 0.05) and more than 1.5-fold changed.

C Genes found to be down-regulated by PARPi as described above (P value < 0.05, 1.5-fold change) in either HT-sensitive cells (left) or CRPC cells (right) were queried against the expression of these genes in the Grasso et al data set in Oncomine. Benign = gray, primary PCa = blue, metastases = orange. Boxplot was generated using the mean expression of the PARPi down-regulated genes in the indicated data sets. Statistical significance determined by two-tailed Student's t-test. Box plots are median and upper and lower quartiles. Whiskers are min and max. For the Grasso et al data set, n = 28 benign prostate tissues, n = 59 localized prostate cancer, and n = 35 metastatic castration resistant.

exogenous expression of E2F1 results in loss of E2F1 regulation by PARP-1. As such, amplified E2F1 may serve as exclusion criteria in future clinical investigation of PARPi in PCa. These data indicate that canonical E2F1 target gene expression is sensitive to PARP-1 function.

To assess the impact of PARP-1 on E2F1 function, chromatin immunoprecipitation (ChIP) analyses were performed. In conditions that were identical to those utilized for the transcriptome analyses in Fig 2, these ChIP analyses indicate that PARP-1 suppression resulted in diminished E2F1 at the E2F1 locus by ~ 40% (Fig 3C, top left). This is important, given that E2F1 is a regulator of *E2F1* gene expression. Additionally, PARP-1 was found at the *E2F1* locus, and PARP-1 residency at this locus was reduced ~ 50% in response to PARPi (Fig 3C, top right). Furthermore, RNA polymerase II residency was reduced by ~ 50%, as was the active transcriptional mark, acetylated histone H4 by ~ 66% (Fig 3C, bottom). These data indicate that PARP-1 enzymatic activity is involved in the biochemical regulation of E2F1 transcriptional function on chromatin.

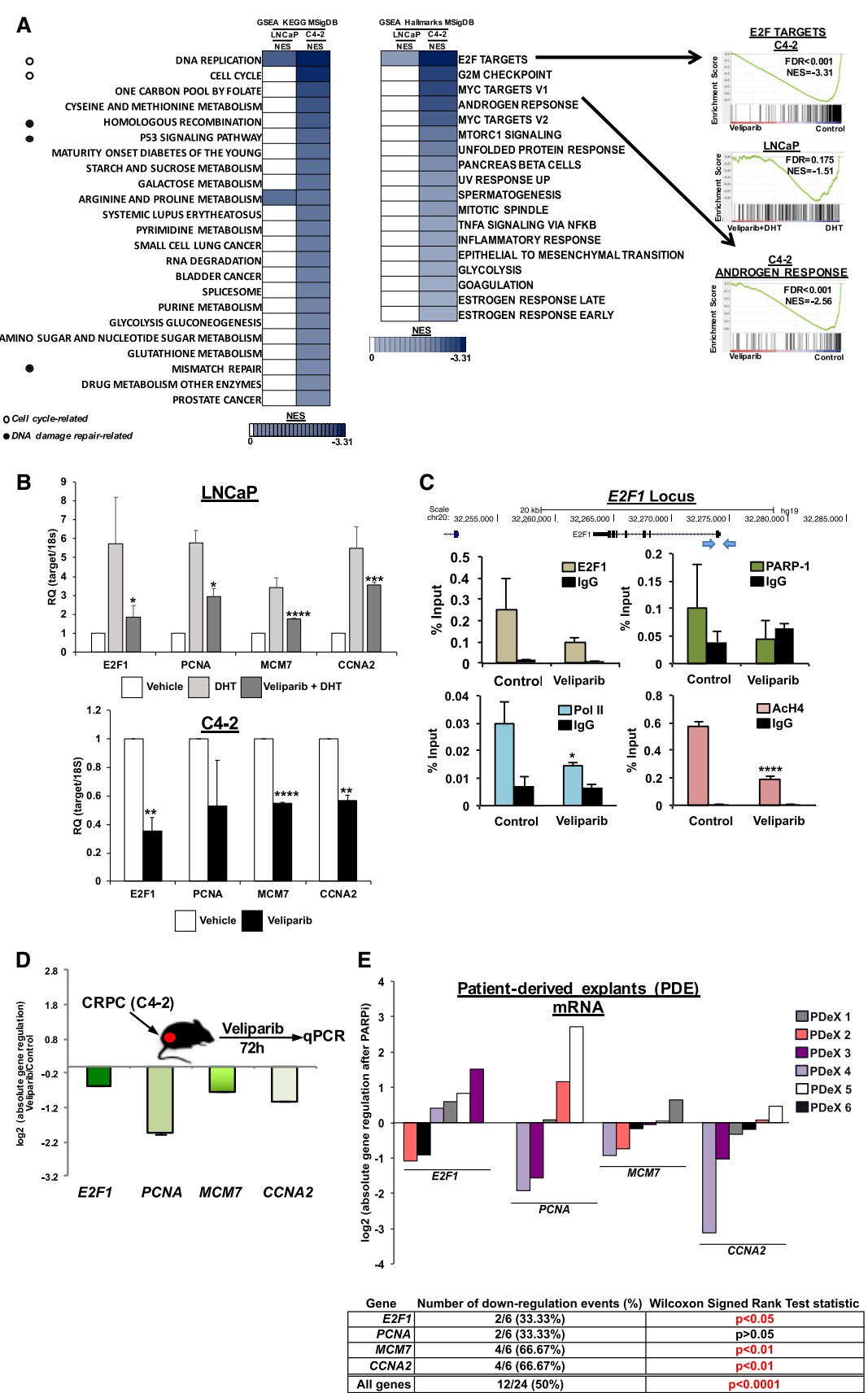

**Figure 3.**

**Figure 3.  PARP-1 regulates pro-oncogenic transcription factor signaling.**

A   Left: Data generated as described above in Fig 2 were utilized for Gene Set Enrichment Analysis (GSEA) Molecular Signature DataBases (MSigDB) KEGG analyses. Cutoff for reporting was a false discovery rate *q* value of < 0.25, and normalized enrichment scores (NES) are shown, with darker colors indicating more enrichment. Middle: Data generated as described above in Fig 2 were utilized for Gene Set Enrichment Analysis (GSEA) Molecular Signature DataBases (MSigDB) KEGG analyses. Cutoff for reporting was a false discovery rate *q* value of < 0.25, and normalized enrichment scores (NES) are shown, with darker colors indicating more enrichment. Open circles indicate cell cycle-related hallmarks, and closed circles indicate DNA damage repair-related hallmarks. Right: Selected GSEA MSigDB Hallmarks pathways are shown with NES and false discovery rate (FDR).

B   Indicated cell lines were treated as depicted in Fig 2. Data are depicted as mean ± standard deviation of three independent biological experiments. Statistical significance was determine by two-tailed Student's *t*-test where *$P$ < 0.05, **$P$ < 0.01, ***$P$ < 0.001, ****$P$ < 0.0001. LNCaP: E2F1, $P$ = 0.0159; PCNA, $P$ = 0.0217; MCM7, $P$ = 4.0936e-6; CCNA2, $P$ = 0.0005. C4-2: E2F1, $P$ = 0.0074; PCNA, $P$ = 0.1258; MCM7, $P$ = 3.7471e-5; CCNA2, $P$ = 0.0031.

C   ChIP-qPCR after C4-2 cells were treated as depicted in Fig 2. Data are depicted as mean ± standard deviation of three independent biological experiments. Statistical significance was determined by two-tailed Student's *t*-test where *$P$ < 0.05, ****$P$ < 0.0001. E2F1 ChIP, $P$ = 0.4610; PARP-1 ChIP, $P$ = 0.1773; Pol II ChIP, $P$ = 0.0305; AcH4 ChIP, $P$ = 7.4261e-5.

D   Athymic nude mice were injected with C4-2 cell mixed with matrigel. Once tumors became 100 mm³, mice were treated with either vehicle control or veliparib. Seventy-two hours later, tumors were harvested, RNA was isolated and used for qPCR quantification of the indicated transcripts. Data are depicted as log2 absolute gene regulation of veliparib samples compared to control samples, ± standard deviation of three independent xenograft tumors.

E   Prostatectomy tissue (*n* = 6) was cultured as previously described, and treated with either vehicle control or veliparib for 6 days. RNA was then harvested from the tissues and used for qPCR quantification of the indicated transcripts. Data are depicted as log2 absolute gene regulation of veliparib samples compared to control samples. Each individual tissue is depicted by a separate bar color. Statistical analyses were performed by Wilcoxon signed rank test.

To assess the impact of PARP-1 on E2F1 function *in vivo*, CRPC (C4-2) xenografts were generated in castrated, immunocompromised mice. Tumor-bearing mice were then treated with the PARPi veliparib for 72 h, sacrificed, and tumors were excised. As shown, the expression of canonical E2F1 target genes (*E2F1*, *PCNA*, *MCM7*, and *CCNA2*) was diminished *in vivo* upon PARP-1 suppression (Fig 3D). To further validate these findings, human tissues were utilized for an explant protocol that has been previously described (Centenera *et al*, 2012, 2013; Schiewer *et al*, 2012; Comstock *et al*, 2013; Goodwin *et al*, 2015; de Leeuw *et al*, 2015; Hartsough *et al*, 2018). Briefly, fresh human PCa samples are obtained at the time of surgical resection, subdivided, and cultured *ex vivo* under conditions that retain the glandular architecture, stromal content, and clinicopathologic features of the original tumor. Explants were exposed to PARPi (or control), and the expression of canonical E2F1 target genes (*E2F1*, *PCNA*, *MCM7*, and *CCNA2*) was assessed. As shown, the response was heterogeneous, but these patient tissues demonstrated significantly diminished E2F1 target gene expression in response to PARPi (Fig 3E). These collective data identify PARP-1 as a major effector of E2F1 function *in vitro*, *in vivo*, and in human PCa tissues.

### PARP-1 effects on E2F signaling are independent of cell cycle phase and distinct from those elicited by CDK4/6 inhibition

To assess the impact of cell cycle phase on PARP-1-mediated E2F1 regulation, HT-sensitive and CRPC cells were treated using conditions identical to those described in Fig 2, and subjected to a BrdU pulse and FACS analyses. As shown, there was no change in DNA replication at an early time point (3 h) or at the time point at which the transcriptional effects of PARP-1 were assessed (16 h; Fig 4A), indicating that cell cycle phase cannot explain the decrease in E2F1 function after PARP-1 suppression, although at later time points, DNA replication is diminished upon PARPi. While E2F1 itself cannot currently be therapeutically targeted, the upstream kinases that positively regulate E2F1 function (cyclin-dependent kinases 4 and 6, CDK4/6) can be inhibited (O'Leary *et al*, 2016), and CDK4/6 inhibitors (CDK4/6i) are under clinical investigation for a number of tumor types, including PCa (NCT02905318, NCT02494921, NCT02555189). The analyses

above indicate that E2F function is under the control of PARP-1, and thus, it was necessary to compare the transcriptional effects of CDK4/6i to PARPi to discern the transcriptional effects of PARP-1. To accomplish this, unbiased transcriptomic data generated in HT-sensitive cells treated with either the CDK4/6i palbociclib or the PARPi veliparib were compared. As shown in Fig 4B, left, there was no significant overlap in the genes up-regulated by CDK4/6i and PARPi (*n* = 1), and minimal overlap in the genes down-regulated by each treatment (*n* = 45). However, these analyses indicate that the genes specifically down-regulated by PARPi were not only the most abundant (*n* = 157), but GSEA MSigDB analyses indicate this gene set was enriched for DNA repair processes, including HR (Fig 4B, right). These data indicate that PARP-1 regulates a cell cycle-independent E2F1 function, distinct from the transcriptional gene regulation by E2F associated with cell cycle control.

### PARP-1 controls of HR factor availability are associated with modulation of the chromatin context of E2F1 function

As the data above identify PARP-1 as a positive regulator of E2F1 activity and subsequent expression of genes controlling HR, the impact of PARP-1 inhibition was compared to that of HR deficiency. Utilizing the HR gene set to generate heatmaps from the unbiased data derived above in Fig 2, it was determined that whether the comparator was DHT in HT-sensitive cells, or vehicle control in CRPC cells, the majority of HR gene expression was diminished with PARPi (Fig 5A, left). In fact, the majority of genes involved in most DNA repair pathways declined after PARPi treatment (Appendix Fig S5). Furthermore, comparison of the unbiased data generated above with a previously developed HR deficiency transcriptional signature (Peng *et al*, 2014) demonstrated a significant overlap in both down-regulated (*n* = 104/151) and up-regulated (*n* = 44/89) genes (Fig 5A, middle). This signature was generated by independently silencing *BRCA1*, *RAD51*, or *BRIT1*, followed by unbiased transcriptomic profiling. The intersection of these conditions serves as the HR deficiency transcriptional signature. This intersection proved to be statistically significant using GSEA analyses (Fig 5A, right). These data suggest PARP-1 suppression reduces availability of HR factors by transcriptional regulation.

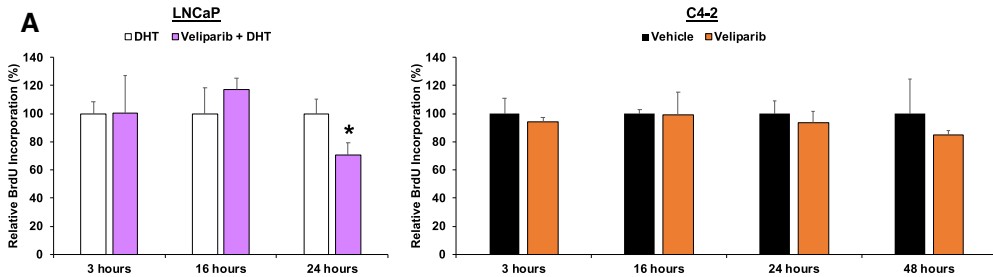

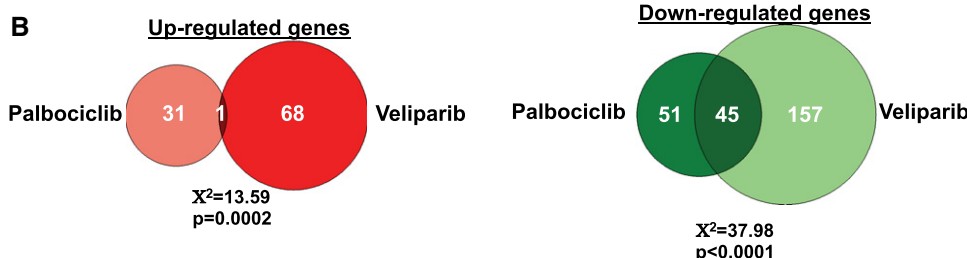

## Up-regulated genes GSEA KEGG

|  | Palbociclib Unique | Union | Veliparib Unique |
|---|---|---|---|
| **Acute myeloid leukemia** | 0.0193 | 1 | 1 |
| **Bladder cancer** | 0.0103 | 1 | 1 |
| **Cell cycle** | 0.000303 | 1 | 1 |
| **Chronic myeloid leukemia** | 0.0236 | 1 | 1 |
| **Colorectal cancer** | 0.0193 | 1 | 1 |
| **Endometrial cancer** | 0.0000108 | 1 | 1 |
| **ErbB signaling pathway** | 0.0298 | 1 | 1 |
| **Focal adhesion** | 0.000862 | 1 | 1 |
| **Glioma** | 0.000862 | 1 | 1 |
| **Melanoma** | 0.000862 | 1 | 1 |
| **Non-small cell lung cancer** | 0.000681 | 1 | 1 |
| **p53 signaling pathway** | 0.0224 | 1 | 1 |
| **Pancreatic cancer** | 0.000862 | 1 | 1 |
| **Pathways in cancer** | 0.00000213 | 1 | 1 |
| **Phosphatidylinositol signaling system** | 0.0241 | 1 | 1 |
| **Prostate cancer** | 4.76E-08 | 1 | 1 |
| **Small cell lung cancer** | 0.00125 | 1 | 1 |
| **TGF-beta signaling pathway** | 0.00125 | 1 | 1 |
| **Thyroid cancer** | 0.0053 | 1 | 1 |

## Down-regulated genes GSEA KEGG

|  | Palbociclib Unique | Union | Veliparib Unique |
|---|---|---|---|
| **Cell cycle** | 0.00000189 | 1.42E-12 | 1.83E-15 |
| **DNA replication** | 0.000000237 | 2.75E-08 | 0.000000033 |
| **Mismatch repair** | 0.00024 | 0.0065 | 0.0273 |
| **Oocyte meiosis** | 0.0224 | 0.000000109 | 0.00437 |
| **Progesterone-mediated oocyte maturation** | 1 | 0.00000122 | 0.000151 |
| **Base excision repair** | 1 | 0.0133 | 0.00325 |
| **Nucleotide excision repair** | 1 | 0.0187 | 0.00437 |
| **Glutathione metabolism** | 1 | 0.0217 | 1 |
| **p53 signaling pathway** | 1 | 0.0000269 | 1 |
| **Pyrimidine metabolism** | 1 | 0.00414 | 0.00325 |
| **Prostate cancer** | 1 | 1 | 0.00325 |
| **Systemic lupus erythematosus** | 1 | 1 | 0.00849 |
| **Homologous recombination** | 1 | 1 | 0.0367 |

**Figure 4.**

    

◀

**Figure 4.  PARP-1 effects on E2F signaling are independent of cell cycle phase and distinct from those elicited by CDK4/6 inhibition.**

A    Indicated cell lines were treated as depicted in Fig 2, and labeled with bromodeoxyuridine (BrdU), harvested at indicated time points and utilized for FACS analyses. Data are depicted as mean ± standard deviation of three independent biological experiments. *$P < 0.05$ as determined by two-tailed Student's *t*-test. LNCaP: 3 h, $P = 0.9838$; 16 h, $P = 0.2197$, 24 h, $P = 0.0207$. C4-2: 3 h, $P = 0.4520$; 16 h, $P = 0.9446$; 24 h, $P = 0.4025$; 48 h, $P = 0.3431$.

B    Top: Data generated as described above in Fig 2 were compared to a separate microarray analysis in which the same cell line was exposed to 1 μM palbociclib instead of veliparib. Cutoffs for comparison were a $P$ value < 0.05, and fold change of 1.5. Venn diagrams shows the overlapping and non-overlapping genes of both down- (top) and up-regulated (bottom) genes in response to either treatment modality. Statistical significance was determined using the Chi-squared statistical test. Bottom: Genes found to be exclusively regulated by palbociclib, commonly regulated by palbociclib and veliparib, or exclusively regulated by veliparib were used for Gene Set Enrichment (GSEA) KEGG pathway analyses. Data indicate both FDR $q$ value, where the darker colors indicate higher confidence (lower $q$). Numbers indicate $q$ values. Blue arrow highlights the Homologous Recombination KEGG pathway.

The impact of PARP-1 activity on the expression of HR genes was validated at the transcript level (~ 20–50% reduction; Fig 5B, left) and at the protein level (~ 15–80% reduction) *in vitro* (Fig 5B right). Validation that these HR genes are E2F1-regulated was accomplished by transiently knocking down E2F1 and examining HR gene expression (Appendix Fig S6A). Transcriptional regulation of HR gene expression was found to be conserved across all PCa/CRPC models tested (Appendix Fig S6B). Furthermore, the dependence of HR gene expression on PARP-1 enzymatic activity was validated *in vivo* (Fig 5C). Additionally, utilizing the patient tissue explant process described in Fig 3 in which prostatectomy tissues are cultured in the laboratory, the reliance of HR gene expression on PARP-1 enzymatic function could be further explored. PARPi thus elicited a more robust and significant decrease of HR gene expression, than canonical E2F1 target genes as described above, but still with patient heterogeneity of response (Fig 5D). Together, these data indicate that PARP-1 inhibition reduces expression of many genes involved in DNA repair (especially HR), suggesting that inhibiting PARP-1 enzymatic function may transcriptionally induce a state of "BRCA-ness", or relative HR deficiency.

To define potential mechanism(s) by which PARP-1 regulates HR gene expression, ChIP-qPCR experiments were performed at regulatory loci of HR genes known to be regulated by E2F1. While there was no clear pattern of altered E2F1 residency at three HR gene loci (*BRCA2*, *RAD51*, and *TOP2A*) after PARPi (Fig 5E, top left graphs), in each case, PARP-1 was found to reside at each locus, and this residency was diminished upon PARPi by ~ 60–83% (Fig 5E, top right graphs). Thus, PARPi destabilizes PARP-1 function at HR gene regulatory loci, likely compromising E2F1 activity. As would be expected, RNA polymerase II and acetylated histone H4 levels were diminished at these HR gene loci in response to PARPi by 40–80 and 28–60%, respectively (Fig 5E, bottom left and right graphs, respectively). Furthermore, it was determined that PARPi alters the activation status the endogenous inhibitor of E2F1 function, the retinoblastoma tumor suppressor (RB; Fig 5F), wherein PARPi resulted in enrichment of hypophosphorylated (active) RB, suggesting that the functions of PARP-1 suppression may be pleiotropic. Additionally, based on the observed decrease in the active acetylated histone H4 mark upon PARPi in Fig 5E, it was determined that this was associated with reduced CBP chromatin occupancy (Fig 5G). These congruous data are important, as CBP is a key histone acetyltransferase with known functions in PCa (Santer *et al*, 2011; Ianculescu *et al*, 2012). Combined, these data indicate that PARP-1 not only resides at HR gene regulatory loci and is diminished upon PARPi, but PARP-1 enzymatic activity appears to support E2F1 in the context of a coactivator, whose functions include modulation of RB.

## Altered HR factor expression is prevalent in human PCa and is enriched during disease progression

Data herein indicate that PARP-1 positively regulates E2F1-mediated HR gene expression in cancer, and that suppression of this activity can potentially induce a "BRCA-ness" phenotype. Given that PARP-1 activity is enhanced as a function of aggressive disease, patterns of HR gene expression were queried in human cancer. An assessment of the TCGA data set (Cancer Genome Atlas Research N, 2015), which includes only primary PCa, demonstrated that when both RNA and DNA alterations are taken in to account, 50.45% of tumors in this data set harbored altered HR gene RNA or DNA (Fig 6A, left). The most frequent HR gene alteration found in primary disease was mRNA up-regulation (65%), while mutations only occurred in 6% (Fig 6A, right). Utilizing the portion of the TCGA data set (Cancer Genome Atlas Research N, 2015) with matched normal and primary PCa tissues, several HR genes were significantly up-regulated in cancer (Fig 6B), suggesting that deregulation of HR gene expression occurs during PCa tumorigenesis. Genes that either did not pass the cutoff for statistical significance or did not increase are shown in Appendix Fig S7. In the PCF-SU2C data set of advanced mCRPC tumors (Robinson *et al*, 2015), there was an increased occurrence of HR gene defects, with 68% of tumors harboring either DNA or mRNA alterations (Fig 6C, left). The most frequent alteration was mRNA up-regulation (71%), while only 10% of these tumors harbored mutations in these HR genes (Fig 6C, right). These observations were supported by two other, independent data sets (Kumar *et al*, 2016; Taylor *et al*, 2010; 94.74% HR gene alteration (Appendix Fig S8, top left), 76% of which was mRNA up-regulation (Appendix Fig S8, top right); 67.65% HR gene alteration (Appendix Fig S8, bottom left), 26% of which was mRNA up-regulation (Appendix Fig S8, top right), respectively). However, the most frequent gene alteration in the second data set was gene amplification, not mutation, further suggesting that HR gene up-regulation is the predominant alteration present in human PCa. Assessment of individual tumor-level data indicates that HR alterations are not mutually exclusive, and the most frequently altered HR gene is *NBN* (22%), while *BRCA1* and *BRCA2* are altered in ~ 7 and 8% of these tumors, respectively (Fig 6D). Several studies have indicated that the frequency of DNA repair gene mutations is elevated in advanced PCa when compared to primary disease (Grasso *et al*, 2012; Robinson *et al*, 2015; Pritchard *et al*, 2016). Data presented herein confirm this and also indicate that HR gene expression is also increased as a function of PCa progression. Combined, these data not only reiterate that HR gene defects occur at a higher frequency in advanced PCa, but the most frequent HR gene aberration is mRNA up-regulation, rather

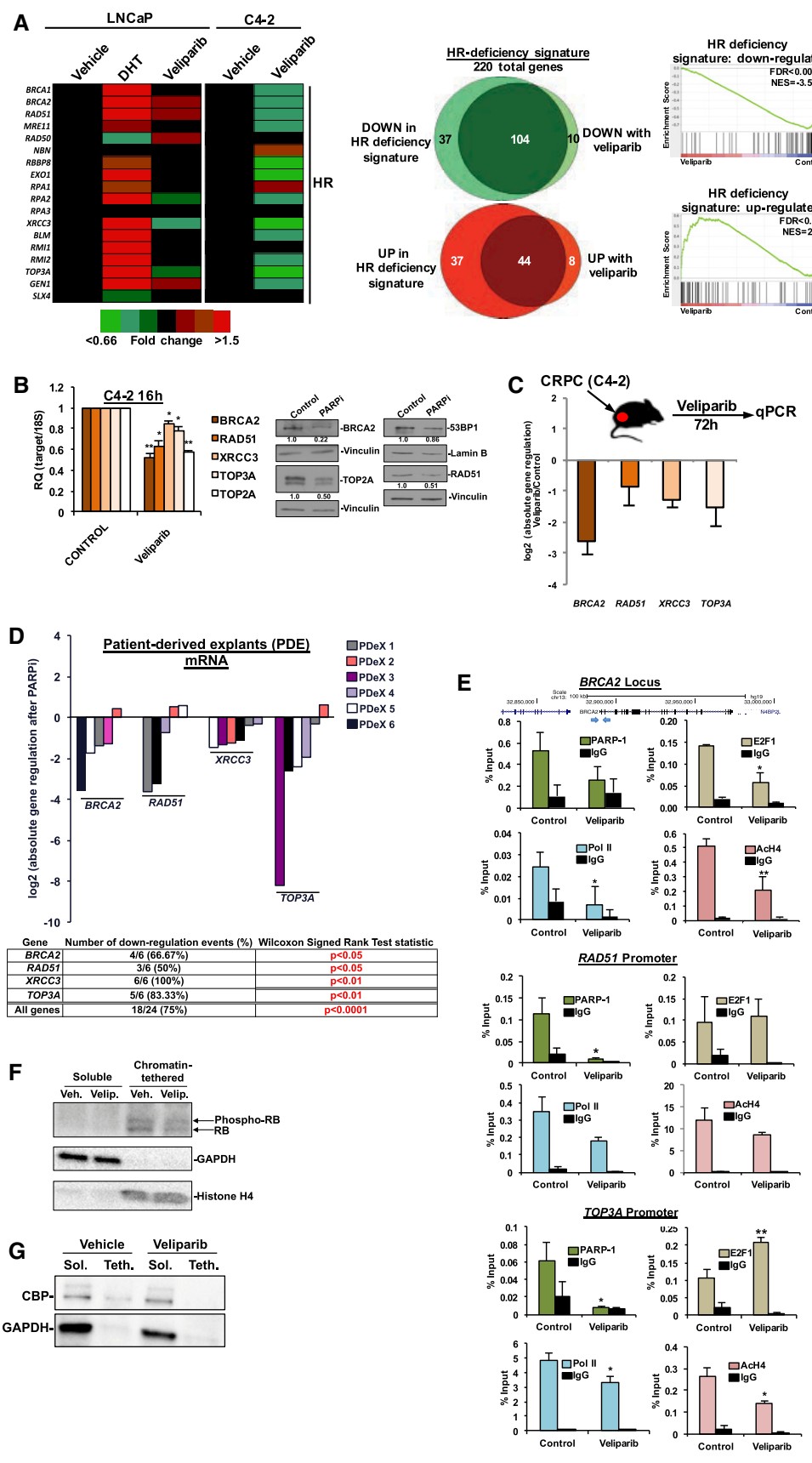

Figure 5.

◄

**Figure 5.  PARP-1 controls of HR factor availability is associated with modulation of the chromatin context of E2F1 function.**

A    Left: Data generated as described above in Fig 2 were used to generate a heatmap of homologous recombination (HR) gene expression after the indicated treatment regimens. Middle: Selected GSEA MSigDB Oncogenic Signature pathways are shown. Right: Data generated as described above in Fig 2 were compared to a previously described HR deficiency transcriptional profile (Peng et al, 2014, Nature Communications). This profile was derived by independently silencing either BRCA1, RAD51, or BRIP1, followed by transcriptional analyses. The union of these three data sets was used to generate the signature. Cutoffs for comparison were a P value < 0.05, and fold change of 1.5. Venn diagrams shows the overlapping and non-overlapping genes of both down- (top) and up-regulated (bottom) genes in the previously defined HR deficiency signature, and the PARPi-responsive transcriptome.

B    Left: C4-2 cells were treated as depicted in Fig 2. Data are depicted as mean ± standard deviation of three independent biological experiments. Statistical significance was determined by two-tailed Student's t-test where *$P < 0.05$, **$P < 0.01$. BRCA2, $P = 0.0046$; RAD51, $P = 0.0151$; XRCC3, $P = 0.0341$; TOP3A, $P = 0.04988$. Right: C4-2 cells were treated as depicted in Fig 2 and immunoblotted with the indicated antisera. Quantifications shown below each band.

C    Athymic nude mice were injected with C4-2 cell mixed with matrigel. Once tumors became 100 mm$^3$, mice were treated with either vehicle control or veliparib. Seventy-two hours later, tumors were harvested, RNA was isolated and used for qPCR quantification of the indicated transcripts. Data are depicted as log2 absolute gene regulation of veliparib samples compared to control samples, ± standard deviation of three independent xenograft tumors.

D    Prostatectomy tissue (n = 6) was cultured as previously described, and treated with either vehicle control or veliparib for 6 days. RNA was then harvested from the tissues and used for qPCR quantification of the indicated transcripts. Data are depicted as log2 absolute gene regulation of veliparib samples compared to control samples. Each individual tissue is depicted by a separate bar color. Statistical analyses were performed by Wilcoxon signed rank test.

E    C4-2 cells were treated as depicted in Fig 2. ChIP was performed using the indicated antisera, and the subsequent DNA was isolated and used in qPCR product using primers designed to amplify the indicated genomic loci: BRCA2 enhancer, RAD51 promoter, or TOP3A promoter. Data are depicted as mean ± standard deviation of three independent biological experiments. Statistical significance was determined by two-tailed Student's t-test where *$P < 0.05$, **$P < 0.01$. BRCA2 locus: E2F1 ChIP, $P = 0.0308$; PARP-1 ChIP, $P = 0.0488$; Pol II ChIP, $P = 0.0471$; AcH4 ChIP, $P = 0.0081$. RAD51 Promoter E2F1 ChIP, $P = 0.7739$; PARP-1 ChIP, $P = 0.0366$; Pol II ChIP, $P = 0.0767$; AcH4 ChIP, $P = 0.1378$. TOP3A promoter: E2F1 ChIP, $P = 0.0074$; PARP-1 ChIP, $P = 0.0500$; Pol II ChIP, $P = 0.0199$; AcH4 ChIP, $P = 0.0158$.

F, G  C4-2 cells treated with 2.5 µM veliparib (Vel.) or vehicle control (Veh.) for 24 h. Cells were then harvested, lysed, and differentially centrifuged as described in the Material and Methods section, resulting in a soluble fraction (Sol.; GAPDH serves as control) or a chromatin-tethered fraction (Teth.; histone H4 serves as control). Immunoblots were performed for the indicated proteins.

Source data are available online for this figure.

than mutation. HR gene defects increase during prostate cancer progression, the most frequent of these defects is mRNA up-regulation. Since the data presented herein demonstrate that HR gene expression is controlled by PARP-1, and that PARP-1 enzymatic activity is increased during prostate cancer progression, there is an association between PARP-1 activity and HR gene expression. These data identify HR gene deregulation as a common feature in advanced disease, further highlighting the potential importance of altered HR gene expression in disease development and/or progression.

**PARP-1 regulates DNA repair factor availability and DNA repair competency**

Based on the fact that PARP-1 transcriptionally regulates HR gene expression, and that the HR gene mRNA up-regulation that frequently occurs in advanced disease is meaningful for the response to PARP-1 inhibitors, it was imperative to assess the impact of exogenous expression of HR factors on functional and biological outcomes after PARP-1 suppression. To accomplish this, multiple model systems were transduced to ectopically express the HR factors BRCA1 and BRCA2, followed by PARP-1 suppression and molecular and cellular readouts as depicted in the schematic in Fig 7A, top. Control transfected cells exhibited reduced cell proliferation in response to PARP-1 suppression (Fig 7A, bottom, white bars). However, these same cell lines first transduced to over-express either BRCA1 or BRCA2 displayed no cell growth inhibition in response to PARP-1 suppression (Fig 7A, bottom, light blue and dark blue bars). To define the potential mechanism underlying this lack of biological response to PARP-1 suppression with BRCA1 or BRCA2 over-expression, cells treated as per Fig 7A for and were utilized for immunofluorescent detection of γH2AX foci as a measurement of DNA DSBs. Control transfected cells treated exhibited an induction of DSBs upon PARP-1 suppression (LNCaP

~ 2-fold; C4-2 ~ 1.5-fold; 22Rv1 ~ 2-fold; Fig 7B, white bars), which was abolished with over-expression of either BRCA1 or BRCA2 (Fig 7B, light blue and dark blue bars). These data indicate that expression dosage of HR factors, which are reduced upon PARP-1 inhibition, has the capacity to alter the biological response to PARP-1 suppression by differential induction of DSBs. While data presented in Fig 1F indicate that the correlation between DSBs and PARP activity is loss during disease progression, data in Fig 7B demonstrate that artificially de-coupling PARP-1 transcriptional regulation of DNA repair factors renders tumor cells unresponsive to PARP inhibition, thus demonstrating that transcriptional regulation of DNA repair factors by PARP-1 has an impact on both the biochemical and the biological response to PARPi.

Combined (as depicted in the schematic in Fig 7C), these analyses reveal that PARP-1 enzymatic and transcriptional functions are elevated as a function of PCa progression, and that the PARP-1-regulated transcriptome includes key oncogenic transcription factors. Furthermore, PARP-1 plays both direct roles in DNA repair and indirect roles through transcriptional regulation of DNA repair gene expression, particularly HR genes. The transcriptional regulation of HR factors is clinically relevant, as the most frequent category of HR gene defects in PCa is mRNA up-regulation, indicating that PARP-1-mediated expression of HR factors holds clinical relevance. Finally, PARP-1-driven expression of HR factors may be a potential determining factor in the anti-cancer effects of PARP-1 suppression through enhancing or inducing BRCA-ness.

# Discussion

Discernment of the molecular mechanisms underlying tumor progression and therapeutic response is critical for the development and proper utilization of treatment strategies in the management of cancer. This study reveals that PARP-1 functions are associated with

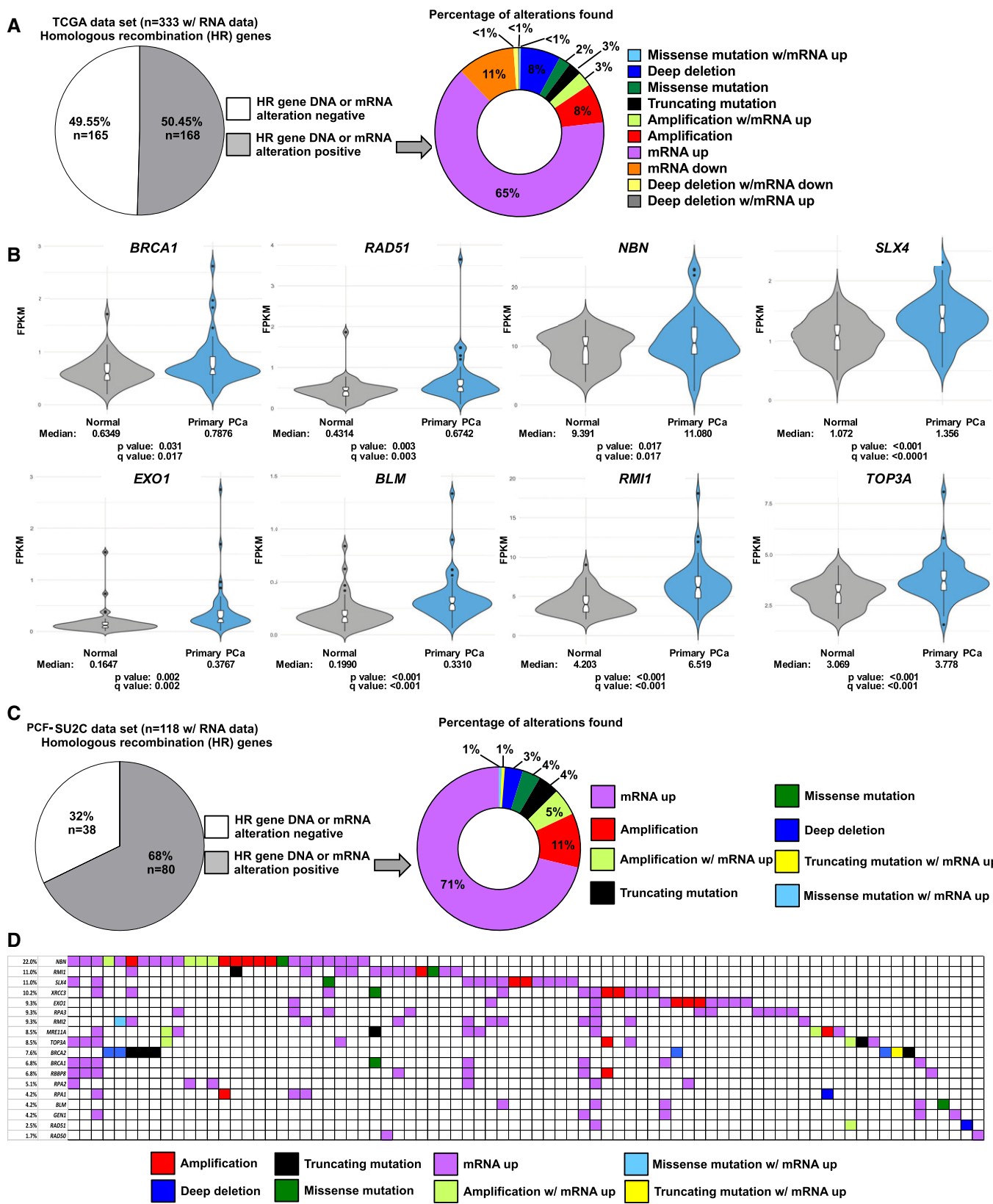

**Figure 6.**

◀

**Figure 6.  Altered HR factor expression is prevalent in human PCa and is enriched during disease progression.**
A  The CBioportal was used to query the DNA and RNA HR gene alterations found in the TCGA primary PCa data set. HR genes queried were *BRCA1, BRCA2, RAD51, MRE11A, RAD50, NBN, RBBP8, EXO1, RPA1, RPA2, RPA3, XRCC3, BLM, RMI1, RMI2, TOP3A, GEN1, SLX4*. Default settings were used.
B  Expression levels of indicated HR pathway genes in primary PCa vs. normal patient samples. Violin plots represent FPKM normalized counts obtained from matched tumor and normal RNA-Seq data from TCGA (*n* = 52) with *P* values generated using paired *t*-tests. Notch is the median, length of notch is 95% confidence interval, and whiskers are 1.5 times the interquartile range from the hinge.
C  The CBioportal was used to query the DNA and RNA HR gene alterations as above using the PCF-SU2C advanced PCa data set.
D  The CBioportal was used to query the DNA and RNA HR gene alterations as above using the PCF-SU2C advanced PCa data set, and the data are presented on a per patient basis.

PCa progression, mediated in part by transcriptional events. Key findings include the following: (i) PARP-1 enzymatic activity is elevated as a function of PCa progression and is associated with poor outcome; (ii) elevated PARP-1 function in advanced CRPC is not associated with increased PARP-1 expression or correlated with DNA DSBs; (iii) PARP-1-regulated transcriptional events are also elevated as a function of PCa progression; (iv) PARP-1 was identified as a major regulator of E2F1 signaling, distinct from those impacted by cell cycle modulation; (v) PARP-1 selectively regulates E2F1-mediated expression of factors governing HR; and (vi) Suppression of PARP-1 can induce BRCA-ness through limiting DNA repair factor availability. Together, these data support a model in which both the enzymatic and transcriptional regulatory function of PARP-1 are elevated as a function of PCa progression to support E2F1-mediated HR gene expression. These studies not only further solidify PARP-1 as a therapeutic target in the management of PCa, but nominate PARP-1 activity as a potential biomarker, and PARP-1 inhibition as a mechanism to induce or enhance BRCA-ness.

Data reported herein indicate that both the enzymatic activity and transcriptional regulatory functions of PARP-1 are elevated as a function of PCa progression. These data are consistent with a previous observation that PARP-1 enzymatic activity is elevated in cell line models of CRPC when compared to HT-sensitive models (Schiewer *et al*, 2012). Additionally, these observations align with studies demonstrating that PARP-1 and PAR are elevated in PCa compared to benign prostatic hyperplasia in a Chinese cohort (Wu *et al*, 2014) and that PARP-1 protein is elevated in cases of primary PCa as compared to normal controls (Salemi *et al*, 2013). In other tumor types, elevated PARP-1 mRNA is associated with poor prognosis in gliomas (Li *et al*, 2016), PARP-1 mRNA is elevated in colon carcinoma when compared to adenoma (Dziaman *et al*, 2014), PARP-1 gene expression is associated with lymph node spread of malignant pleural mesothelioma (Walter *et al*, 2016), and PARP-1 mRNA and protein are elevated in endometrial adenocarcinoma (Bi *et al*, 2013). Both PARP-1 mRNA and protein are highly expressed in small cell lung cancer (Byers *et al*, 2012), but PARP-1 protein has been shown to associate with longer PFS in limited-stage small cell lung cancer (Kim *et al*, 2014). High PARP-1 protein is associated with shorter survival in soft tissue sarcomas (Kim *et al*, 2016), poor prognosis in gastric cancer (Park *et al*, 2015); is an independent prognostic factor for decreased PFS and OS in high-grade serous ovarian carcinoma (Gan *et al*, 2013); is associated with higher grade, ER negativity, and TNBC, as well disease-free and overall survival in operable invasive BrCa (Rojo *et al*, 2012); and is associated with poor prognosis in oral squamous cell carcinoma (Mascolo *et al*, 2012). Additionally, PARP-1 protein is higher in triple negative breast cancer (TNBC) specimens than in non-TNBC breast cancers, and high PARP-1 expression is associated with worse PFS in TNBC

(Zhai *et al*, 2015). Combined, these studies indicate that elevated PARP-1 occurs in many tumor types, and may have prognostic value. Data shown herein provide some of the first evidence that PARP-1 hyperactivation is associated with disease progression, independent of DNA damage markers.

The underlying mechanisms that lead to elevated PARP-1 function in CRPC do not appear to be associated with elevated DNA DSBs or increased PARP-1 protein expression, and as such, efforts are ongoing to determine the molecular drivers and biological consequence of elevated PARP-1 enzymatic activity in CRPC. One clue may lie in the observation that castration alters not only PAR levels, but also NAD$^+$ and other PAR-related metabolites in murine kidneys (Gartemann *et al*, 1981). Interestingly, high PARP activity is associated with platinum sensitivity and improved PFS in epithelial ovarian cancer (EOS; Veskimae *et al*, 2016), and PARP-1 positivity is associated with higher grade and complete response to first-line chemotherapy in EOS (Godoy *et al*, 2011), further suggesting that assessing PARP-1 activity has potential as a meaningful biomarker. The underlying mechanisms that drive heightened PARP-1 activity as a function of PCa progression may be due to deregulated NAD$^+$ production, since NAD$^+$ is the substrate for PARP-1 production. It has previously been reported that transcriptional regulation by PARP-1 is affected by recruitment of an NAD$^+$ synthase enzyme (NMNAT-1; Zhang *et al*, 2012). However, there are several other enzyme involved in NAD$^+$ production, and each demonstrate some patient-derived alterations in human malignancy. There are also unexplored patient-derived alterations in PARP-1 itself, which may affect PARP-1 activity. Furthermore, poly(ADP-ribose) glycohydrolase (PARG), which hydrolyzes PAR moieties, harbors patient-derived alterations of unknown relevance, which may impact PARP-1 activation status by differentially regulating PAR levels. It has recently been reported that PARG impacts the response to PARPi in models of pancreatic cancer (Chand *et al*, 2017). Irrespective of the mechanism(s) by which PARP-1 is hyperactivated in advanced PCa, studies described herein yield novel insights into the downstream functions of elevated PARP-1 activity.

While the expression and enzymatic activity of PARP-1 are altered in several tumor types, delineation of the transcriptional targets of PARP-1 in PCa models revealed that not only is HR gene expression is regulated by PARP-1 activity, the expression of HR genes is elevated during prostate transformation. These data suggest that PARP-1-mediated HR gene expression may promote aggressive phenotypes. Conversely, PARP-1 inhibitors may induce BRCA-ness (in HR-competent tumors) or enhance BRCA-ness in HR-deficient tumors. This is consistent with a previous report that demonstrated that TGFβ signaling in wild-type *BRCA1/2* breast cancers down-regulates HR gene expression, and renders breast cancer cells more sensitive to PARPi (Liu *et al*, 2014). There is also evidence that *BRCA2* can

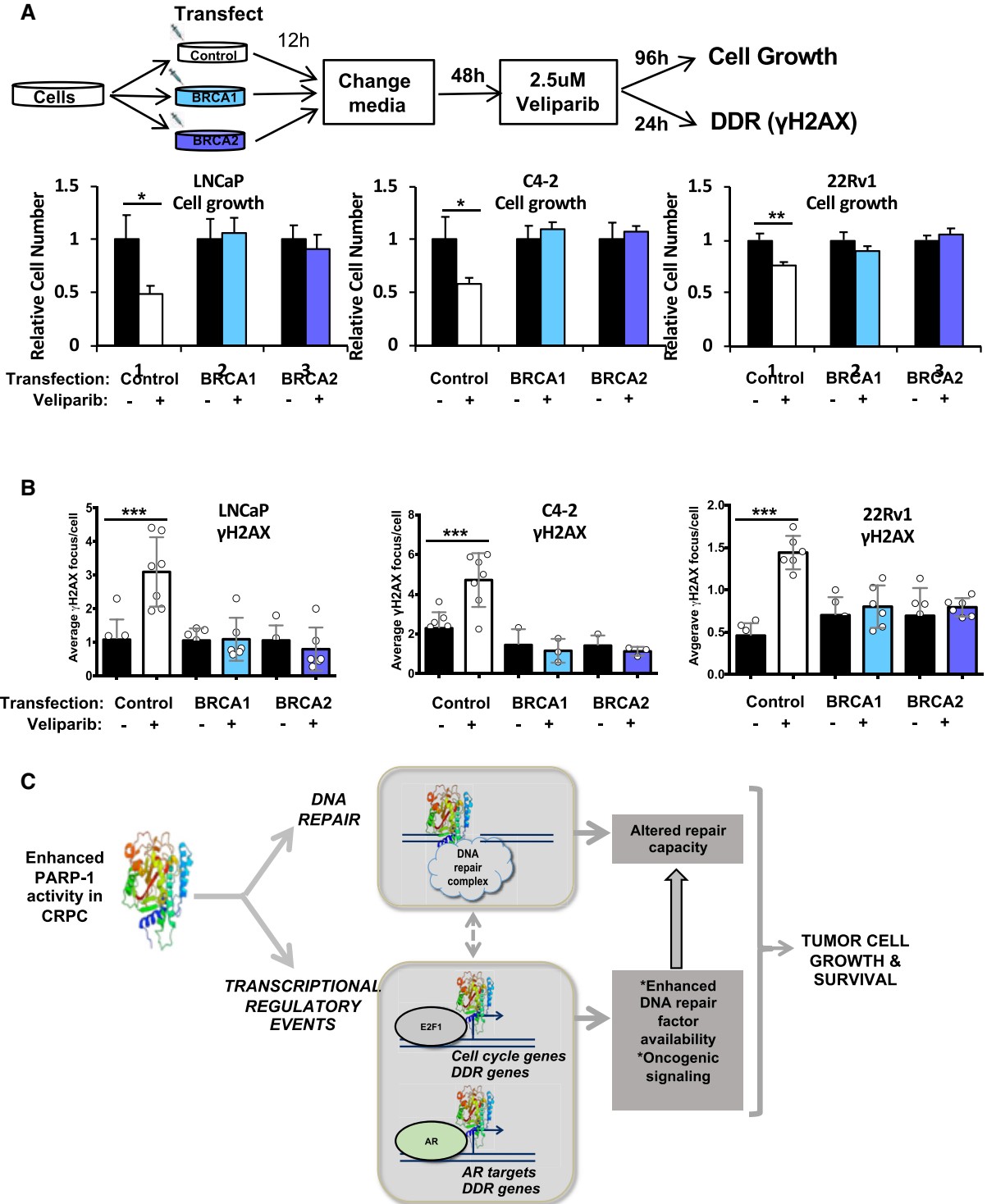

**Figure 7. PARP-1 regulates DNA repair factor availability and DNA repair competency.**

A, B (A) Indicated cell lines were transfected with indicated constructs and treated with veliparib. Cell growth and (B) DDR via γH2AX was assessed. Data represent median ± standard deviation of independent biological replicates. Control transfected and vehicle control treated cells are set to 1. *P value < 0.05, **P value < 0.01, ***P < 0.001. Statistical significance was determined by two-tailed Student's t-test. LNCaP cell growth: Control transfection, P = 0.0220; BRCA1 transfection, P = 0.67787; BRCA2 transfection, P = 0.4676. C4-2 cell growth: Control transfection, P = 0.0354; BRCA1 transfection, P = 0.1638; BRCA2 transfection, P = 0.2519. 22Rv1 cell growth: Control transfection, P = 0.0039; BRCA1 transfection, P = 0.1085; BRCA2 transfection, P = 0.2781. LNCaP γH2AX: Control transfection, P = 0.0008; BRCA1 transfection, P = 0.9035; BRCA2 transfection, P = 0.4685. C4-2 γH2AX: Control transfection, P = 0.0009; BRCA1 transfection, P = 0.6362; BRCA2 transfection, P = 0.4217. 22Rv1 γH2AX: Control transfection, P < 0.0001; BRCA1 transfection, P = 0.4698; BRCA2 transfection, P = 0.4937.

C Graphical abstract of data presented herein. TF = transcription factor.

be post-transcriptionally regulated in PCa by the lncRNA *PCAT-1* (Prensner *et al*, 2014). Another pharmacological approach to generating BRCA-ness through transcriptional regulation has been reported through use of histone deacetylase inhibitors in breast cancer cells (Wiegmans *et al*, 2015), which has also been examined in PCa (Chao & Goodman, 2014). Furthermore, in response to hypoxia, a dynamic E2F switch occurs, in which E2F1 is replaced by E2F4 at the *BRCA1* promoter, thus causing gene repression and transcriptionally regulated BRCA deficiency (Bindra *et al*, 2005). Irrespective of the mechanism that leads to reduced HR gene expression, means to accomplish this may be of benefit given the frequency with which these genes are up-regulated at the mRNA level as a function of PCa progression, and PARP-1 suppression appears to be capable of significantly limiting HR gene expression in *BRCA* wild-type PCa. Whether this is unique to either PCa in specifically or *BRCA1/2* wild-type tumor cells in general is an area of active interest.

Identification of PARP-1 as a regulator of E2F1 transcriptional function in PCa, specifically with regard to regulation of HR gene expression, sheds new light as to the molecular impact of PARP-1 function in cancer. PARP-1 regulation of E2F1 function is consistent with previous studies which demonstrate that PARP-1 regulates E2F1 transcriptional activity with respect to driving cellular proliferation (Simbulan-Rosenthal *et al*, 1998, 2003; Leger *et al*, 2016). Furthermore, PARP-1 has been reported to be involved in the regulation of E2F1-induced apoptosis (Kumari *et al*, 2015). Data presented herein demonstrate that PARP-1 resides at regulatory loci of E2F1 target HR genes, and blocking PARP-1 enzymatic activity consistently reduced PARP-1 residency consistently at each target locus investigated. The effect of PARP-1 function on E2F1 residency appeared to be context-dependent, but at each locus investigated, the recruitment of RNA polymerase II and enrichment of an epigenetic marker of active transcription (acetylated histone H4) were dependent upon PARP-1 enzymatic activity. Furthermore, PARP-1 suppression appears to regulate RB activity, either directly or indirectly, which may contribute to E2F1 modulation. Ongoing studies are have been designed to investigate the mechanisms by which PARP-1 impinges upon the transcriptional repressive functions of RB. These data suggest that PARP-1 functions to regulate a permissive chromatin state for transcriptional activation of HR genes by E2F1. This is likely based on the chromatin compaction/relaxation capacity of PARP-1 function, and subsequent function of epigenetic writers/readers, such as histone acetyltransferases. Future studies are designed to investigate the mechanisms by which PARP-1 regulates E2F1-driven transcriptional activation in PCa.

Finally, findings herein provide insights into novel biomarkers of potential clinical use in PCa, which is of critical importance given the lack of clinical biomarkers with utility in predicting PCa progression or therapeutic response, and the clinical data that indicate that PARPi responsiveness is not necessarily linked to HR status (Fong *et al*, 2009; Audeh *et al*, 2010; Gelmon *et al*, 2011; Kaye *et al*, 2012; Sandhu *et al*, 2013; Coleman *et al*, 2015; Mirza *et al*, 2016), including in PCa (Mateo *et al*, 2015; Clarke *et al*, 2018). This held true in the Phase III NOVA trial (Mirza *et al*, 2016), in which patients with platinum-sensitive recurrent ovarian cancer receiving niraparib (a PARPi) maintenance therapy had increased progression-free survival (PFS) compared to placebo control, irrespective of *BRCA1/2* mutational or HR deficiency status. Analyses of clinical samples demonstrated that PARP-1 enzymatic activity is elevated as a

function of PCa progression, and that high PARP-1 activity strongly correlated with decreased progression-free survival, implicated PARP-1 as a driver of lethal malignant phenotypes. Strikingly, this elevation in PARP-1 enzymatic activity in advanced CRPC was not associated with either higher expression of PARP-1 itself, or with increased evidence of DNA DSBs, which are known to activate PARP-1 enzymatic function, implying tumor cells may select for higher PARP-1 function through other mechanisms. Regardless, studies herein suggest that PARP-1 enzymatic output may be a novel biomarker of PCa aggressiveness or potential to progress to CRPC. Furthermore, defining the PARP-1-dependent transcriptome in PCa models revealed that the targets of PARP-1 transcriptional regulation, including HR genes, are also elevated as a function of PCa progression in clinical data sets. These data suggest that a transcriptional profile of PARP-1 effectors has the potential to be a biomarker of PCa progression. Current investigation into whether this transcriptional profile, or PARP-1 enzymatic output, has utility in predicting therapeutic response is ongoing. While PARPi is in clinical development for PCa management, the clinical value of targeting PARP-1 for the prevention of CRPC development, and progression in other tumor types, should be evaluated.

In sum, the studies herein reveal fundamental new knowledge of PARP-1 function in malignancy. The data presented are impactful in cancer, as PARP-1 activity is increased as a function of disease progression and is associated with poor outcomes. These novel findings have the potential to impact cancer therapy, based on the discovery that PARP-1 suppression has the capacity to induce or enhance BRCA-ness through regulation of DNA repair factor availability.

# Materials and Methods

### Standard immunohistochemistry

Tissue microarrays (TMAs) of primary PCa were provided by Dr. Kelly (TJU), and the TMAs of mCRPC were provided by Dr. Visakorpi (U. Tampere). TMAs were deparaffinized in xylene, washed in decreasing quantities of EtOH, followed by a water wash. Antigen retrieval was done in sodium citrate buffer with boiling. Endogenous peroxidase was blocked using $H_2O_2$, background was blocked with mouse serum, and tissues were covered in a 1:500 dilution of mouse monoclonal anti-PAR antibody (Trevigen, Gaithersburg, MD, USA) then incubated overnight at 4°C. Slides were then washed with PBS and developed using the Vectastain Elite ABC Mouse IgG Kit (Vector Laboratories, Burlingame, CA, USA) according to manufacturer's specifications and the Liquid DAB Substrate Kit (Invitrogen, Carlsbad, CA, USA) according to manufacturer's specifications. Slides were then counterstained using hematoxylin by standard methods and washed in increasing EtOH concentrations followed by xylene, and then, coverslips were mounted. Slides were then scored blindly for both PAR intensity and PAR percent positivity by a board-certified pathologist (Dr. Parsons, TJU).

### Multiplexed, fluorescent immunohistochemistry

Tissue microarray slides were stained using the OPAL™ multiplex fluorescent staining system from PerkinElmer (PerkinElmer cat. no.

NEL794B001KT). Immunofluorescent detection of pγH2AX(Ser139; CST #2577) was carried out with the first using a 1:200 dilution, followed by PARP-1 (Active Motif #39559) using a 1:100 dilution and PAR (Trevigen Inc., 4335-AMC-050) using a 1:200 dilution. The TMA slide was first blocked with 3% $H_2O_2$ for 10 min, then treated with animal-free protein blocker (Vector Laboratories cat. no. SP-5030) for 15 min, and then incubated overnight at 4°C with the pγH2AX primary antibody diluted in Antibody Dilution Buffer (Ventana Medical Systems cat. no. ADB250). The next day, the TMA slide was incubated with EnVision+ System—HRP labeled polymer goat anti-rabbit secondary antibody (Dako cat. no. K4003) for 30 min at room temperature followed by incubation with OPAL-FITC fluorophore for 10 min.

Next, the slide was loaded onto the Ventana autostainer using the Ventana reagents for the machine. The pγH2AX antibody was completely removed using heat retrieval with CC2 buffer, only leaving the FITC fluorophore behind that was crosslinked to the tissue. The PARP-1 antibody was applied manually, followed by manual application of the OPLA-Cy3 reagent. Next, the PARP-1 antibody was completely removed from the slide, leaving the Cy3 fluorophore behind as it was crosslinked to the tissue. The final incubation occurred with the PAR antibody and the OPAL-Cy5 fluorophore. The slide was incubated with DAPI, washed, and coverslipped using prolong gold as the mounting medium. No cross-reactivity in signals was observed between antibodies, demonstrating that the removal of the antibodies between staining cycles was complete.

Individual cores were imaged on the Vectra™ 2 quantitative slide imaging system. Non-neoplastic and cancer areas were annotated by a pathologist resulting in 156 non-neoplastic areas, 277 primary cancer areas, and 159 mCRPC areas. Missing cores and cores without glands were excluded from the annotation. The InForm™ software was used to obtain the gray-level staining images of individual fluorophores. The amount of nuclear staining in individual nuclei was measured for all four fluorophores (DAPI, FITC, Cy3, Cy5), and intensity levels were normalized across the four TMA slides. Normalized intensities of each fluorophore were dichotomized into positive or negative using as a cutoff the median intensity across all nuclei within the TMA. The percent of positive nuclei for every antibody was counted in benign and neoplastic glands. Alternatively, the average expression of each fluorophore across all nuclei in each annotated region was determined.

## Cell culture and treatments

LNCaP and C4-2 cells were maintained in minimum essential media (IMEM) supplemented with 5% FBS (heat-inactivated fetal bovine serum). 22Rv1 cells were maintained in Dulbecco's modified Eagle's media supplemented with 10% FBS. All media were supplemented with 2 mmol/l of L-glutamine and 100 units/ml penicillin-streptomycin. Veliparib was obtained from Enzo Life Sciences (Farmingdale, NY, USA) and dissolved in DMSO and used at indicated concentrations. For steroid-depleted conditions, cells were plated in appropriate phenol red-free media supplemented with 5 or 10% charcoal dextran-treated FBS (CDT) as appropriate. DHT was dissolved in EtOH and used at indicated concentrations. Cell lines were not cultured for longer than 6 months after receipt from their original source, or no longer than 45 passages. Cell lines are authenticated by ATCC annually.

## Microarray analysis

Cells were seeded at equal density in steroid-depleted (CDT) conditions then treated as indicated with as specified for 16 h; RNA was isolated using TRIzol (Invitrogen) according to manufacturer's specifications, and submitted for microarray analysis to the Sidney Kimmel Cancer Center Cancer Genomics Shared Resource. Gene expression was profiled using the Affymetrix Human Gene 1.0 ST microarray (Santa Clara, CA, USA), with hybridization performed using the GeneChip Hybridization Oven 645, followed by scanning on Affymetrix Gene Chip Scanner 3000. Data preprocessing was performed in Affymetrix Expression Console 1.1 using iterPLIER summarization with PM-GCBG background correction and quantile normalization.

## Gene expression analysis

Cells were seeded at equal density in steroid-depleted (CDT) conditions and were treated as specified; RNA was isolated using TRIzol and cDNA generated using SuperScript III (Invitrogen). Quantitative PCR was conducted with primers described in Appendix Table S1 and with an ABI StepOne machine and PowerSybr in accordance with the manufacturer's specifications.

## ChIP analysis

Cells were cultured in media containing CDT for 72 h and treated as indicated. ChIP analyses and qPCR were conducted as previously described (60), using primers described in Appendix Table S1.

## Xenograft analysis

Four-week-old male BALB/c nu/nu mice were purchased from Charles River, Inc. C4-2 ($2 \times 10^6$ cells) were resuspended in 100 µl of saline with 50% Matrigel (BD Biosciences) and were implanted subcutaneously into the flank of the mice. All tumors were staged for 4 weeks before starting the drug treatment. For assessment of *in vivo* gene expression, tumors from mice were treated with a single dose of veliparib (100 mg/kg via oral gavage) and harvested 72 h after treatment. Tissue was harvested at indicated after 6 days RNA was isolated using TRIzol. No statistical methods were used for animal sample size estimate, and no blinding was done. Animals were randomized into the two treatment regimens via coin flip. Mice were housed in standard conditions. All animal work was done in compliance with the regulations set forth by the Jefferson University IACUC.

## Chromatin tethering assays

C4-2 cells were treated with either 2.5uM veliparib or vehicle control, then harvested and processed 24 h later as previously described (Schiewer *et al*, 2012).

## Human prostate tumor *ex vivo* culture

Human prostate *ex vivo* explant cultures were conducted as previously described (de Leeuw *et al*, 2015). Briefly, fresh tissue was obtained from a pathologist immediately following radical

**The paper explained**

**Problem**

While the roles poly(ADP-ribose) polymerase-1 (PARP-1) performs in response to DNA damage are increasingly well understood, as are the roles of PARP-1 in other aspects of genome integrity (telomeric maintenance and replication fork stability), the other chromatin-associated function of PARP-1 (transcription) has not been fully explored as a means to regulate DNA repair.

**Results**

*Central findings are as follows*: (i) PARP-1 enzymatic activity is increased as a function of disease progression and is associated with poor outcome. (ii) Elevated PARP-1 enzymatic function in advanced disease is not attributable to increased DNA DSB repair. (iii) Identification of the PARP-1-regulated transcriptome reveals relevance for disease progression. (iv) PARP-1 regulates pro-oncogenic transcription factor signaling, including E2F1. (v) PARP-1 effects on E2F signaling are independent of cell cycle phase and distinct from those elicited by CDK4/6 inhibition. (vi) PARP-1 regulates homologous recombination (HR) factor availability via modulating chromatin at E2F1 binding sites. (vi) Altered HR factor expression is prevalent in human PCa and is enriched during disease progression. (viii) PARP-1 regulates DNA repair factor availability and DNA repair competency.

**Impact**

These data establish three essential points. *First*, they provide the first evidence that PARP-1 enzymatic and transcriptional functions are elevated as a function of disease progression, irrespective of DNA repair. *Second*, the data establish an unexpected role for PARP-1 in controlling DNA repair gene expression, and reveal a new paradigm for PARP-1 to function as an enhancer or inducer of "BRCA-ness". *Finally*, these data demonstrate the clinical relevance of PARP-1-regulated E2F1-driven expression of HR factors and provide striking new evidence for novel biomarkers of human disease.

prostatectomy. The de-identified specimens were processed under a laminar flow hood, using sterile technique, and transported to the lab in IMEM on ice. The Thomas Jefferson University Institutional Review Board has reviewed this procurement protocol and determined this research to be in compliance with federal regulations governing research on de-identified specimens and/or clinical data [45 CFR 46.102(f)]. The following procedures were conducted under sterile tissue-culture conditions. Veterinary dental sponges (Novartis Cat. #96002) were placed in 12-well plates and soaked in 500 ml media (IMEM supplemented with 5% heat-inactivated FBS, hydrocortisone, insulin from bovine pancreas, and 100 units/ml penicillin-streptomycin) and appropriate treatment (either vehicle control or 2.5 μM veliparib) for 5–10 min at 37°C. Tissue was placed into the lid of a 10-cm plate and dissected into 1-mm$^3$ pieces with a scalpel. Three pieces of tissue were placed on each sponge, using sterile tweezers or forceps. Plates were placed in an incubator at 37°C and 5% $CO_2$. Media were replaced every day with appropriate treatment. Tissue was harvested at indicated after 6 days RNA was isolated using TRIzol.

While there was no clinical investigation reported in this study, informed consent was obtained from all subjects and that the experiments conformed to the principles set out in the WMA Declaration of Helsinki and the Department of Health and Human Services Belmont Report.

## Cell growth assays

Cells were seeded at equal densities, treated as indicated, and harvested at 96 h. At the time of harvest, cell number was determined using trypan blue exclusion and a hemocytometer.

## Antibodies and immunoblotting

Protein isolation and immunoblotting were conducted as previously described (Knudsen *et al*, 1998), using antisera described in Appendix Table S1.

## Data availability

The data sets produced in this study are available in the following databases: Microarray data: Gene Expression Omnibus GSE118222 (https://www.ncbi.nlm.nih.gov/geo/query/acc.cgi?acc=GSE118222).

**Expanded View** for this article is available online.

## Acknowledgements

We gratefully thank all the members of the Knudsen laboratory for their intellectual and technical support. Additionally, we thank the following institutions that supported this work: the NIH/NCI grants to KEK (R01 CA176401, R01 CA182569, R01 CA217329, P30 CA056036) and the Sidney Kimmel Cancer Center (5P30CA056036), the Prostate Cancer Foundation (to MJS and KEK), and the Translational Pathology and MetaOmics core facilities at SKCC.

## Author contributions

Conceptualization: MJS and KEK; methodology: MJS, BK, and KEK; investigation and data analysis: MJS, ACM, NG, FH, SG, RdL, SGZ, JE, SH, TP, RB, PMcC, CMcN, SNC, YC-F, PG, JJMcC, NPN, AAS, ED, LJB; biostatistics: BEL; bioinformatics: CMcN; writing—original draft: MJS and KEK; writing—review and editing: all authors; funding acquisition: MJS and KEK; resources: TV, GVR, CDL, EJT, LGG, APD, WKK, FYF. All authors read and approved the manuscript.

## Conflict of interest

The authors declare that they have no conflict of interest.

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
