## [Review Process File · EMBO Molecular Medicine]

PARP-1 regulates DNA repair factor availability

Matthew J. Schiewer, Amy C. Mandigo, Nicolas Gordon, Fangjin Huang, Sanchaika Gaur, Renée de Leeuw, Shuang G. Zhao, Joseph Evans, Sumin Han, Theodore Parsons, Ruth Birbe, Peter McCue, Christopher McNair, Saswati N. Chand, Ylenia Cendon-Florez, Peter Gallagher, Jennifer J. McCann, Neermala Poudel Neupane, Ayesha A. Shafi, Emanuela Dylgieri, Lucas J. Brand, Tapio Visakorpi, Ganesh V. Raj, Costas D. Lallas, Edouard J. Trabulsi, Leonard G. Gomella, Adam P. Dicker, Wm. Kevin Kelly, Benjamin E. Leiby, Beatrice Knudsen, Felix Y. Feng, and Karen E. Knudsen

Review timeline:

Submission date:	22 December 2017
Editorial Decision:	28 February 2018
Revision received:	09 July 2018
Editorial Decision:	17 September 2018
Revision received:	10 October 2018
Accepted:	25 October 2018

Editor: Céline Carret

Transaction Report:

1st Editorial Decision

28 February 2018

Thank you for the submission of your manuscript to EMBO Molecular Medicine. We have now heard back from the three referees whom we asked to evaluate your manuscript.

You will see from the comments below that all referees find the study potentially interesting. Referee 1 has questions and constructive suggestions to make the paper more focused and compelling. Referee 2 agrees with referee 1 that the introduction should be rewritten and centered on prostate cancer only. On a similar idea, deriving a general signature doesn't seem necessary and ref. 2 suggests removing this data. Finally, referee 3 comments on further analyses to be done to increase conclusiveness and insights.

We would therefore welcome the submission of a revised version within three months for further consideration and would like to encourage you to address all the criticisms raised as suggested to improve conclusiveness and clarity. Please note that EMBO Molecular Medicine strongly supports a single round of revision and that, as acceptance or rejection of the manuscript will depend on another round of review, your responses should be as complete as possible.

I look forward to receiving your revised manuscript.

***** Reviewer's comments *****

Referee #1 (Remarks for Author):

In this manuscript, Schiewer and colleagues study the role of PARP1 in transcriptional regulation in prostate cancer, identifying E2F1 as a key regulator in this process. More specifically, many factors involved in HR are directly under control of E2F1, in which PARP1 inhibition induces a 'BRCAness' like phenotype in these cells. In general, I find this a good paper with interesting results. I would however ask the authors to reposition the introduction of the paper, as an attempt is made to generalise their findings to other tumor types, using prostate cancer 'as a model'. I believe the authors should limit the paper to the actual tumor type that was studied, as generalisation is not illustrated.

In addition, I have the following comments, see below. This is quite a long list, but I would like to stress that I am very interested and positive about the paper.

major comments

1. In the kaplan-meier plot (Fig. 1C) only survival is shown for mCRPC, combining the different groups. Please show the groups separately as well (albeit that the groups get very small), and also include the same curve for primary
2. How does the γ H2AX and PAR intensity with values between 0-150% reflect the values (-2 - +2) in figure 1G? please explain the difference in units used on both Y axis
3. I would suggest also use a cut-off for the effect size in the RNA-seq analysis. (Fig. 2A and 2B). In addition to p value cut-off, also use a fold-change cut-off.
4. What is the overlap of significant genes between LNCaP cells and C4-2 cells? What is the overlap of genes affected by DHT/veh versus DHT/veliparib? Please include Venn diagrams depicting overlap of all three conditions.
5. Does C4-2 cells have increased parp-1 enzymatic activity as compared to LNCaP(as resistant tissue show higher PAR activity as compared to sensitive cells/tissue)?
6. For sensitive and resistant conditions in Figure 2, only 1 cell line is being used for each condition. Additional validation cell lines should be included for validation purposes.
7. In the legends of Figure 2C, it is indicated that all genes with p value cutoff <0.05 and fold change of 1.5 were further used for analyses. However, for the LncAP conditions in Figure 2A, this would indicate that hardly any genes would be left. Please provide a list of these genes as a supplementary table, indicating all genes used in the visualizations of Figures 2 and 3.
8. In Figure 3A, the NES scores of C4-2 cells are quite impressive, but those for LNCAP are not. Even androgen response for LncAP is not significant, and hardly any genesets show any effect in LNCAP. Please explain.
9. Figure B3: please include a vehicle controle for the LNCAP conditions. Are the genes downregulated after PARPi normally under control of DHT (as compared to vehicle)?
10. For the genes selected as E2F1-responsive in Figure 3D and E, please provide ChIP-QPCR data (or publicly available E2F1 ChIP-seq data in LncAP?) to indicate that these genes are indeed under regular control of E2F1. Alternatively, please provide expression data of these genes upoin siE2F1.
11. As E2F1 regulates its own expression, can the effects of PARPi on E2F1 levels be uncoupled from the biological activity of E2F1 as transcriptional regulator? The authors could exogenously introduce E2F1 into these cells, followed by PARPi. Are the 4 genes affected in Figure 3 still downregulated with exogenous E2F1 present?
12. The absence of proof is no proof of absence; in Figure 4A the authors conclude that no effect of PARPi is found on BrDU incorporation, even though cell cycle genesets were found in Figure 3. Is the timepoint chosen sufficient to see an effect? What if the authors would have incubated the cells longer?
13. How can the number of genes in Figure 4B be different from that as depicted and used in Figure 2? Furthermore, the authors state in the text that the genes shared by Palbociclib and Veliparib-treated cells are 'merely' 45 genes, but these are in fact close to 50% of all genes downregulated by

palbociclib treatment and should therefore not be excluded. With that, the KEGG pathways of the other subgroups in this analysis should be provided as well.

14. Even though the title of Figure 5 stated that PARP1 controls HR factor availability through E2F1, the causal positioning of E2F1 in the results shown in Figure 5 is not strong. How can the authors claim causal involvement of E2F1 in these findings, as 'no clear pattern of E2F1 residency' was found?

15. Figure 5A is completely focused on HR genes (which are under control of AR in this setting). How does this deviate from all other AR-responsive genes in their data? Is this different, or merely a general behavior of a very specific subset of genes, in relation to all other DHT-affected genes?

16. How can the results in Figure 7B be reconciled with the findings in Figure 1F? The cell line results don't seem to be in line with the clinical data.

17. Transcription rewiring and the E2F1 role therein (depicted in Figure 7C) is not convincingly shown. Furthermore, the gene expression data of Figure 5A shows that HR genes are under direct control of DHT, while the figure now shows that HR genes are NOT under control of AR. How could this be?

Minor comments

1. In the following sentence the word SENSITIVE might be missing; hormone therapy (HT) SENSITIVE-PCA.

'Furthermore, PARPi has generated promising clinical trial data in advanced PCa. Initially, human tissues from primary, hormone therapy (HT)-PCa and metastatic CRPC (mCRPC) were queried for PARP-1 enzymatic activity via immunohistochemistry (IHC) for PAR (Poly(ADP)-ribose, the product of PARP-1 enzymatic activity) (Figure 1A)'.

2. In the following sentence:

'These data indicate that PARP-1 enzymatic function is not ONLY elevated in CRPC, but also predictive of PFS, which is associated with disease specific mortality.' , the word ONLY is missing

3. Please add a color scale for the heatmaps in Fig. 2C and Supplementary Fig. 2. Does it represent z-score or log₂ expression values? Also, please include y axis label for figure 2C.

4. Fig 6 in legend it says Fig 8

Referee #2 (Remarks for Author):

This manuscript from Schiewer and co-workers address the mechanistic role of PARP-1 in advancement of prostate cancer. They demonstrate increased PARP activity in the course of disease progression and that PARP activity promotes E2F1-mediated induction of DNA repair factors involved in HR. Overall, the manuscript is somewhat descriptive in nature on the role of PARP1 in transcriptional regulation and would benefit from a more mechanistic approach. The introduction of an unnecessary and vague signature of PARP-1 activity is confusing.

Specific comment:

In the introduction the authors argue that the lack of response in some BRCA cancers to PARP inhibitors call for more mechanistic understanding on PARP to understand the reason. There are numerous studies focussed on PARPi resistance that involve reactivation of HR or short wiring the DDR network. This is completely ignored. The suggestion is to change the introduction and focus more on PARP - HR and Prostate Cancer.

The authors describe a novel signature of PARP-1 activity. What the authors demonstrate are DHT-induced transcripts differentially affected by a PARPi in the context of their cell line. Extraction of a signature applicable widely seems difficult and unnecessary. It is unclear exactly what this signature looks like and how it can be applied to identify PARP-1 activity, please remove.

The authors identify that HR gene expression alters in the course of disease progression based on TCGA data. They also identify mutations in HR genes. These data are descriptive and do not tell about the HR status during PCa development. Indeed, these authors have demonstrated that HR status is highly important to response to PARP inhibitors. Here, the authors should query how the HR function is affected during disease progression.

Minor comment:

Abbreviation DHT should be explained first time it is used - dihydrotestosterone (DHT)

Referee #3 (Remarks for Author):

The manuscript presents a remarkable body of experimental work, supported by solid in-silico analysis of own and publicly available omics data. Furthermore, the results presented appear strong, thus making them of likely interest to the research community.

I would only like to comment on few aspects that I think could be improved, detailed below.

* While representing a remarkable body of work, and understanding the difficulty in working with PDX models and their inherent heterogeneity, the results of Fig 3 D and E are somewhat underwhelming. Please clarify whether the changes in D are statistically significant. Also, I'm not convinced that the 'majority' of changes in E are in the 'right' direction. Perhaps the author can elaborate on how they reached that conclusion.

* In Fig. 4.B, left, the overlap in the down-regulated genes appears highly significant. Thus, it would be important to clarify whether the enrichment shown in Fig 4.B, right, is based on standard GSEA, or it's based on the hyper-enrichment test (i.e., the one available through GSEA here: <http://software.broadinstitute.org/gsea/msigdb/annotate.jsp>). I suspect it is the latter, since the authors include the title "Veliparib down-regulated only." If the former, however, it would be necessary, and reassuring, to show that GSEA on the Palbociclib down signature didn't return HR, etc. among the enriched genesets.

* In Fig. 6B, if those reported are nominal p-values, it's inappropriate, and FDR q-values should be reported instead. Clearly, some of the q-values will not be significant. The authors can still make the case for an "HR enrichment" by testing the HR genes as a set by GSEA.

1st Revision - authors' response

09 July 2018

Thank you for the opportunity to address the positive critiques of the reviewers. We have thoroughly addressed each comment as described below in blue font. The accompanying new data/subsequent analyses have both been included in the revised manuscript and have been embedded into the body of the letter. We look forward to potential publication in *EMBO Molecular Medicine*.

Referee #1:

We thank reviewer 1 for their enthusiasm for the study, and the thoughtful critiques provided. We have addressed each concern to completion, as described below, which has improved this study.

1. In the Kaplan-Meier plot (Fig. 1C) only survival is shown for mCRPC, combining the different groups. Please show the groups separately as well (albeit that the groups get very small), and also include the same curve for primary.

After re-analyzing the data with an expert biostatistician (Dr. Leiby), the IHC data derived from the CRPC TMAs have been additionally assessed using a manual score, defining quartiles, and presenting the KM curves of these quartiles. These data indicate that tumors with the highest level of PARP-1 activity are associated with significantly shorter progression-free survival than as compared to with the lowest level of PARP-1 activity. As shown below, these data are now provided as Figure 1C in the revised manuscript.

The reviewer's suggestion to investigate the impact of PARP-1 enzymatic activity on progression-free survival in prostate cancer patients was an excellent idea. The primary PCa and mCRPC data are derived from different cohorts, and unfortunately we are unable to complete these analyses, as the majority of the primary PCa patients in this cohort have yet to relapse, making this type of analysis impossible.

2. How does the γ H2AX and PAR intensity with values between 0-150% reflect the values (-2 - +2) in figure 1G? please explain the difference in units used on both Y axis

We apologize for any confusion in data presentation. Panels 1D-F are representative of percent positivity, and panel G is intensity. We have analyzed the data as percent positive (with a median intensity cut-off) for all panels, and the figure has been adjusted accordingly (revised Figure 1G). We have also included these data below for ease of review. These data confirm using yet another analytic approach that PARP-1 enzymatic activity (as measured by PAR) is associated with DSBs (as measured by γ H2AX) in non-neoplastic and primary PCa tissues. However, this association is lost in mCRPC tissues, indicating that PARP-1 enzymatic activity is uncoupled from DSBs in PCa progression.

Revised Figure 1G. Correlation between γ H2AX and PARP-1 enzymatic activity is lost in mCRPC. Spearman correlation test between PAR and γ H2AX (% positive with a median intensity cut-off).

3. I would suggest also use a cut-off for the effect size in the RNA-seq analysis. (Fig. 2A and 2B). In addition to p value cut-off, also use a fold-change cut-off.

For downstream analyses, fold-change cut-off was used. In revised Figure 2A and 2B, data were replaced with graphs that depict both significance and fold-change cut-offs, as shown below. While this does not change the original interpretation of the data (that PARP inhibition results in altered transcriptional profiles in prostate cancer cells), these revised graphs better reflect the data and statistical analyses thereof. We thank the reviewer for this suggestion.

Revised Figures 2A and 2B. Identification of the PARP-1-regulated transcriptome. Volcano plots of transcripts found to be differentially regulated by DHT v. EtOH (left), DHT v. PARPi followed by DHT (middle) in LNCaP cells or PARPi v. Vehicle in C4-2 cell (right). Red dots indicate transcripts that were both statistically significantly altered ($p < 0.05$) and more than 1.5 fold-changed.

4. What is the overlap of significant genes between LNCaP cells and C4-2 cells? What is the overlap of genes affected by DHT/veh versus DHT/veliparib? Please include Venn diagrams depicting overlap of all three conditions.

We have included a Venn diagram (as shown below, left; revised Appendix Figure S2B) to indicate the overlap of differentially regulated genes (not transcripts) as requested, and the data indicate there are both overlapping and distinct transcriptional changes elicited by each condition and in the individual cell lines. These data indicate that there may be a core transcriptional program regulated by PARP-1 in prostate cancer cells, which includes a large number of DHT-responsive genes (n=169), but the transition to castration resistance likely expands the relevance of PARP-1 regulated transcription, given the larger number of transcripts that are altered upon PARPi (n=1810 unique genes regulated by PARP-1).

Revised Appendix Figure S2B. Overlap of PARP-1 regulated transcriptomes in HT-sensitive and CRPC models. Genes found to be significantly different $p < 0.05$ and > 1.5 fold change) in the indicated cell lines under the conditions described are depicted. DHT=dihydrotestosterone.

5. Does C4-2 cells have increased parp-1 enzymatic activity as compared to LNCaP (as resistant tissue show higher PAR activity as compared to sensitive cells/tissue)?

We have previously published that C4-2 cells have increased PARP-1 enzymatic activity than LNCaP cells (Schiewer *et al.*, *Cancer Discovery* 2012). This is now appropriately discussed and cited on page 5.

6. For sensitive and resistant conditions in Figure 2, only 1 cell line is being used for each condition. Additional validation cell lines should be included for validation purposes.

We have performed validation studies in VCaP, 22Rv1, and LNCaP-abl models. As shown below, and in revised Appendix Figure S6B, these data demonstrate the reduction of HR gene expression upon PARPi is conserved across all PCa/CRPC models tested.

Revised Appendix Figure S6B. PARPi reduces HR gene expression in multiple PCa models. Indicated cell lines were treated as depicted in Figure 2. Data are depicted as mean \pm standard deviation of at least three independent biological experiments. Statistical significance was determined by Student's *t* test where $*=p < 0.05$, $**=p < 0.01$, $****=p < 0.0001$.

7. In the legends of Figure 2C, it is indicated that all genes with p value cutoff < 0.05 and fold change of 1.5 were further used for analyses. However, for the LNCaP conditions in Figure 2A, this would indicate that hardly any genes would be left. Please provide a list of these genes as a supplementary table, indicating all genes used in the visualizations of Figures 2 and 3.

We apologize for any misunderstanding. As stated above in response to comment 3, revised Figures 2A and 2B depict both significance and fold-change cut-offs. We have also included gene lists in Appendix Table S1. As stated above in response to comment 4, PARP-1 regulates a core set of genes in prostate cancer cells and alters the expression of a majority of DHT-responsive genes.

Furthermore, there is a larger number of genes that are under regulation of PARP-1, demonstrating that the PARP-1-regulated transcriptome is expanded upon the transition to castration resistance.

8. In Figure 3A, the NES scores of C4-2 cells are quite impressive, but those for LNCaP are not. Even androgen response for LNCaP is not significant, and hardly any genesets show any effect in LNCaP. Please explain.

The reduced number of NES scores reported within the LNCaP data was due to the stringency of cut-offs utilized. The NES for Androgen Response in LNCaP was -0.71, however the FDR q-value is 0.976, which does not satisfy the pre-set significance threshold. Additionally, as shown in revised Appendix Figure S2B, as well as modified Figure 3B, top, DHT induced gene regulation is often negated by PARPi, which may explain the lack of Androgen Response in the LNCaP context. Other pathways present within the C4-2 data were present in the LNCaP data, and we have reported those that pass the statistical threshold pre-set for significance. We utilized the statistical cut-off recommended by the GSEA developers. Only data that passed significance were included. Additionally, as shown in the Venn diagram in response to critique #4, there are more genes that are differentially expressed in the CRPC cell line, which may contribute to the statistical observations seen for these data sets.

9. Figure B3: please include a vehicle control for the LNCaP conditions. Are the genes downregulated after PARPi normally under control of DHT (as compared to vehicle)?

We have included the vehicle control conditions for LNCaP in panel 3B as requested. As shown below and in revised Figure 3B, each canonical E2F1 target gene is DHT responsive, and in each case, the induction of these genes by DHT is diminished by PARPi. These data indicate, when combined with the C4-2 data, that both mitogen-stimulated and basal E2F1 target gene expression is reduced by PARPi.

Revised Figure 3B. PARPi reduces the mitogen-induced expression of canonical E2F1 target gene expression. Indicated cell lines were treated as depicted in Figure 2. Data are depicted as mean +/- standard deviation of at least three independent biological experiments. Statistical significance was determined by Student's *t* test where **p*<0.05, ***=*p*<0.001, ****=*p*<0.0001.

10. For the genes selected as E2F1-responsive in Figure 3D and E, please provide ChIP-QPCR data (or publicly available E2F1 ChIP-seq data in LNCaP?) to indicate that these genes are indeed under regular control of E2F1. Alternatively, please provide expression data of these genes upon siE2F1. We have included E2F1 ChIP-seq tracks in multiple cell lines (publicly available: LNCaP-abl, LM2 (breast cancer cell line), LNCaP, and; LNCaP from our lab) at these canonical E2F1 target gene loci below for referee review only (Chong *et al.*, *Nature* 2009; Wenzel *et al.*, *Dev Biol* 2011; Costa *et al.*, *Oncogene* 2013; Thwaites *et al.*, *Mol Cell Biol* 2014; Hallstrom *et al.*, *Cancer Cell* 2008; Santos *et al.*, *Cancer Res* 2014; Lin *et al.*, *PLoS One* 2013). These data demonstrate that robust E2F1 localization at each locus in each cell line queried.

Furthermore, expression of these genes was queried upon transient E2F1 knockdown, and these data demonstrate that the expression of these canonical E2F target genes is reduced upon E2F1 knockdown. These data are shown below and also included in revised Appendix Figure S4A.

Left: for reviewer only. ChIP-seq tracks demonstrate that E2F1 resides at canonical E2F1 target gene promoters in multiple models.

Right: Revised Appendix Figure S4A. E2F1 knockdown reduces canonical E2F1 target gene expression. C4-2 cells were transfected with either scrambled siRNA (siControl) or siRNA targeted to E2F1 (siE2F1). Data are depicted as mean \pm standard deviation of at least three independent biological experiments. Statistical significance was determined by Student's *t* test where *= $p < 0.05$, **= $p < 0.01$, ****= $p < 0.0001$.

11. As E2F1 regulates its own expression, can the effects of PARP1i on E2F1 levels be uncoupled from the biological activity of E2F1 as transcriptional regulator? The authors could exogenously introduce E2F1 into these cells, followed by PARPi. Are the 4 genes affected in Figure 3 still downregulated with exogenous E2F1 present?

Our data during initial submission demonstrated that E2F1 self-regulation was sensitive to PARPi, and the reviewer suggests an interesting experiment. As shown below and in revised Appendix Figure S4B, models of exogenous E2F1 were generated, and E2F1 target gene expression was examined after PARP inhibition. The data demonstrate that with exogenous expression of E2F1, E2F1 target gene expression is no longer under the control of PARP-1. These data indicate that exogenous expression of E2F1 results in loss of E2F1 regulation by PARP-1. As such, amplified E2F1 may serve as exclusion criteria in future clinical investigation of PARPi in PCA.

Revised Appendix Figure S4B. Exogenous expression of E2F1 uncouples PARP-1 regulation of E2F1 regulation of canonical E2F1 target genes. C4-2 cells were infected with either a control GFP-encoding adenovirus (AdGFP) or an E2F1-encoding adenovirus (AdE2F1). Left: Exogenous E2F1 expression was validated via qPCR. Right: Cells infected as described were either treated with vehicle control or 2.5 μ M veliparib, and cell cycle gene expression was determined via qPCR.

12. The absence of proof is no proof of absence; in Figure 4A the authors conclude that no effect of PARPi is found on BrdU incorporation, even though cell cycle genesets were found in Figure 3. Is the timepoint chosen sufficient to see an effect? What if the authors would have incubated the cells longer?

We have now included later timepoints for the BrdU incorporation experiments in revised Figure 4A, and below. These data indicate that while the transcriptional changes seen at 16 hours (when unbiased transcriptomics approaches were utilized) are not associated with altered DNA replication, and thus not due to cell cycle position, the biological outcome of these transcriptional changes is indeed diminished DNA replication.

Revised Figure 4A. Kinetics of PARPi effects on DNA replication. Indicated cell lines were treated as depicted in Figure 2, and labeled with bromodeoxyuridine (BrdU), harvested at indicated time points and utilized for FACS analyses. Data are depicted as mean \pm standard deviation of at least three independent biological experiments. *= $p < 0.05$ as determined by Student's t test.

13. How can the number of genes in Figure 4B be different from that as depicted and used in Figure 2? Furthermore, the authors state in the text that the genes shared by Palbociclib and Veliparib-treated cells are 'merely' 45 genes, but these are in fact close to 50% of all genes downregulated by palbociclib treatment and should therefore not be excluded. With that, the KEGG pathways of the other subgroups in this analysis should be provided as well.

We apologize for the confusion. In figure 2, the data depicted are at the transcript level. Not all of the transcripts identified as being PARPi responsive are protein-coding. Figure 4B depicts gene expression data. We have removed the word 'merely' from the text. We have also conducted statistical analyses of these Venn diagrams after further analysis by our expert statistical team. Using Chi-squared statistics, it was determined that the genes down-regulated by palbociclib and veliparib differ from each other in a statistically different manner ($X^2=37.98$, $p < 0.0001$), which held true for genes up-regulated by either treatment ($X^2=13.59$, $p=0.0002$). We have included these statistical analyses in the revised manuscript as revised Figure 4B.

KEGG pathway analyses for the other subgroups has been added to revised Figure 4B, and is shown below. These data indicate while there are some overlapping gene sets that are down-regulated by both CDK4/6 inhibition and PARPi, homologous recombination is uniquely down-regulated in response to PARPi. Furthermore, there is no overlap between the gene sets up-regulated by CDK4/6 inhibition and PARPi. As such, the original conclusion that HR gene expression is uniquely responsive to PARPi (and not CDK4/6i) is confirmed.

Revised Figure 4B. CDK4/6i and PARPi affect both similar and distinct pathways.

Genes found to be exclusively regulated by palbociclib, commonly regulated by palbociclib and veliparib, or exclusively regulated by veliparib were used for Gene Set Enrichment (GSEA) KEGG pathway analyses. Data indicate both FDR q value, where the darker colors indicates higher confidence (lower q). Blue arrow highlights the Homologous Recombination KEGG pathway.

14. Even though the title of Figure 5 stated that PARP1 controls HR factor availability through E2F1, the causal positioning of E2F1 in the results shown in Figure 5 is not strong. How can the authors claim causal involvement of E2F1 in these findings, as 'no clear pattern of E2F1 residency' was found?

The HR factors queried are known E2F1 target genes (Tategu *et al.*, *Gene Regul Syst Bio* 2007; Iwanaga *et al.*, *Oncogene* 2004; Kachhap *et al.*, *PLoS One* 2010; Stevens *et al.*, *DNA Repair (Amst)* 2004; Biswas *et al.*, *Cancer Res* 2012). We have included E2F1 ChIP-seq tracks in multiple cell lines (publicly available: LNCaP-abl, LM2 (breast cancer cell line), LNCaP, and; LNCaP from our lab) at these HR gene loci below for referee review only.

For reviewer only. ChIP-seq tracks demonstrate that E2F1 resides at HR gene loci in multiple models.

Furthermore, E2F1 was knocked down and the impact on HR gene expression was examined via qPCR. As shown below and in revised Appendix Figure S6A, the data indicate that reducing E2F1 levels results in diminished expression of these HR gene mRNAs, validating that they are under E2F1-driven regulation.

Revised Appendix Figure S6A. Knockdown of E2F1 reduces HR gene expression. C4-2 cells were transfected with either scrambled siRNA (siControl) or siRNA targeted to E2F1 (siE2F1). Data are depicted as mean +/- standard deviation of at least three independent biological experiments. Statistical significance was determined by Student's *t* test where * $p < 0.05$, ** $p < 0.01$, **** $p < 0.0001$.

Additionally, a recently published manuscript (Komori *et al.*, *Scientific Reports*) demonstrated that deregulated E2F1 function is not necessarily directly linked to elevated E2F1 binding at target gene promoters.

Finally, as detailed below in response to Reviewer #2, Comment #1, we have conducted studies that have determined that PARP-1 enzymatic activity regulates chromatin occupancy of a key histone acetyltransferase (CBP). Data in the revised manuscript also demonstrate that PARP-1 impinges upon the function of the endogenous inhibitor of E2F1 function (RB, retinoblastoma tumor suppressor). As such, we have modified the language associated with Figure 5.

15. Figure 5A is completely focused on HR genes (which are under control of AR in this setting). How does this deviate from all other AR-responsive genes in their data? Is this different, or merely a general behavior of a very specific subset of genes, in relation to all other DHT-affected genes? This appears to hold true for most of the DDR genes that are DHT-responsive as indicated in original Supplemental Figure 3 (now revised Appendix Figure S5), but not all DHT-response transcripts, as shown in Appendix Table S1. This is true for both AR/DHT-repressed as well as AR/DHT-induced gene expression programs.

Furthermore, using a previously characterized set of AR/DHT-responsive target genes, the majority of these genes are oppositely regulated by DHT and PARPi in LNCaP. These data demonstrate that, while the Androgen Response Hallmark does not pass statistical cut-off parameters (as discussed above in response to Comment 8), both the gene repression and induction in response to DHT stimulation are modulated by PARP inhibition. Additionally, this AR/DHT-responsive gene set is modulated by PARPi in CRPC cells in the absence of hormonal stimulation (as shown below and in revised Appendix Figure S2A). Together, these data indicate that PARP activity modulates a significant number of DHT-responsive genes, which includes both DNA repair genes, and previously defined direct AR target genes. Furthermore, the data derived from C4-2 cells indicate that a large number of genes are under the control of PARP-1, irrespective of DHT stimulation.

Revised Appendix Figure S2A. PARPi impinges on AR transcriptional activity. Previously defined androgen/AR regulated genes were used to examine the effect of PARPi in the data generated in Figure 2.

16. How can the results in Figure 7B be reconciled with the findings in Figure 1F? The cell line results don't seem to be in line with the clinical data.

We apologize for any confusion regarding data interpretation. Data in Figure 1F indicate that the correlation between DSBs and PARP activity is lost during disease progression. Data in Figure 7B demonstrates that artificially de-coupling PARP-1 transcriptional regulation of DNA repair factors renders tumor cells unresponsive to PARP inhibition, thus demonstrating that transcriptional regulation of DNA repair factors by PARP-1 has an impact on both the biochemical and the biological response to PARPi. We have modified the text on page 13 to better reflect our interpretation of the data.

17. Transcription rewiring and the E2F1 role therein (depicted in Figure 7C) is not convincingly shown. Furthermore, the gene expression data of Figure 5A shows that HR genes are under direct

control of DhT, while the figure now shows that HR genes are NOT under control of AR. How could this be?

We have adjusted the depiction based on the reviewer's comments and thank them for their insightful suggestion.

Minor comments

1. In the following sentence the word SENSITIVE might be missing;hormone therapy (HT) SENSITIVE-PCA. 'Furthermore, PARPi has generated promising clinical trial data in advanced PCa. Initially, human tissues from primary, hormone therapy (HT)-PCa and metastatic CRPC (mCRPC) were queried for PARP-1 enzymatic activity via immunohistochemistry (IHC) for PAR (Poly(ADP)- ribose, the product of PARP-1 enzymatic activity) (Figure 1A)'.
We have adjusted the text on page 5. We thank the reviewer for the thoughtful edit.

2. In the following sentence: 'These data indicate that PARP-1 enzymatic function is not ONLY elevated in CRPC, but also predictive of PFS, which is associated with disease specific mortality.', the word ONLY is missing
We apologize for this oversight and have adjusted the text to state page 5.

3. Please add a color scale for the heatmaps in Fig. 2C and Supplementary Fig. 2. Does it represent z-score or log2 expression values? Also, please include y axis label for figure 2C.
Color scale bars have been included and represent Z-scores. We have also included a y axis label for the box and whisker plots in Figure 2C.

4. Fig 6 in legend it says Fig 8
We have adjusted the figure legend and appreciate the reviewer's critique.

Referee #2:

We thank reviewer 2 for positive comments and for the suggestions, which were addressed to completion as described below:

1. Overall, the manuscript is somewhat descriptive in nature on the role of PARP1 in transcriptional regulation and would benefit from a more mechanistic approach.
Data presented in the original submission demonstrated that PARP-1 regulates the local chromatin environment by diminishing RNA pol II and acetylated histone H4 at HR gene regulatory loci, thus leading to diminished E2F1-driven HR gene expression. We have conducted chromatin tethering assays for CBP, a key histone acetyltransferase. Now included as revised Figure 5G, these data suggest that PARPi reduces the chromatin occupancy of a key histone acetyltransferase, which is congruous with diminished acetylated histone H4 at E2F1 target gene regulatory loci.

Revised Figure 5G. PARPi reduces the chromatin residency of CBP, a key histone acetyltransferase. C4-2 cells treated with 2.5uM veliparib (Vel.) or vehicle control (Veh.) for 24 hours. Cells were then harvested, lysed, and differentially centrifuged as described in the material and methods section, resulting in a soluble fraction (Sol.) (GAPDH serves as control) or a chromatin-tethered fraction (Teth.) (histone H4 serves as control). Immunoblots were performed for the indicated proteins. A representative image of at least three independent experiments is shown.

We have also determined that PARPi alters the activation status the endogenous inhibitor of E2F1 function, the retinoblastoma tumor suppressor (RB). Phosphorylated RB is the inactive form, which releases E2F1 to regulate transcriptional activation. As shown below, and in revised Figure 5F, indicate PARP-1 enzymatic activity regulates the capacity of RB to repress E2F1 function. Future studies will be directed at discerning how PARP-1 modulates RB, either directly or indirectly.

Revised Figure 5F. PARPi impinges upon RB, the endogenous inhibitor of E2F1. C4-2 cells treated with 2.5 μ M veliparib (Vel.) or vehicle control (Veh.) for 24 hours. Cells were then harvested, lysed, and differentially centrifuged as described in the material and methods section, resulting in a soluble fraction (Sol.) (GAPDH serves as control) or a chromatin-tethered fraction (Teth.) (histone H4 serves as control). Immunoblots were performed for the indicated proteins. A representative image of at least three independent experiments is shown.

Combined, these data support the conclusion that PARP-1 is a key regulator of the local chromatin molecular environment of E2F1 sites of function. These data have been added as revised Figure 5F and 5G.

2. The introduction of an unnecessary and vague signature of PARP-1 activity is confusing. We have revised the text, and no longer use the term signature. However, we feel that the PARP-1 regulated transcriptome identified in cell lines increases as a function of disease progression, coupled with the elevation of PARP-1 enzymatic activity (as depicted in Figure 1), strongly suggests that PARP-1 functions are associated with prostate cancer progression. As such, we have opted to keep these data in the manuscript.

3. In the introduction the authors argue that the lack of response in some BRCA cancers to PARP inhibitors call for more mechanistic understanding on PARP to understand the reason. There are numerous studies focussed on PARPi resistance that involve reactivation of HR or short wiring the DDR network. This is completely ignored. The suggestion is to change the introduction and focus more on PARP - HR and Prostate Cancer.

We have rewritten the introduction to include references to PARPi resistance and DDR network rewiring. We have also focused the introduction more on PARP/HR/prostate cancer, including the newly published phase II study combining PARPi and AR-directed therapy in men with advanced prostate cancer which showed clinical benefit, irrespective of HR status (Clarke *et al.*, *Lancet Oncology* 2018).

4. The authors describe a novel signature of PARP-1 activity. What the authors demonstrate are DHT-induced transcripts differentially affected by a PARPi in the context of their cell line. Extraction of a signature applicable widely seems difficult and unnecessary. It is unclear exactly what this signature looks like and how it can be applied to identify PARP-1 activity, please remove. As stated above in response to comment 2, we are no longer referring to the transcriptomic changes induced by PARPi in our cellular models as a signature. However, the genes down-regulated in both the context of liganded AR (LNCaP model), as well as CRPC cells in the absence of androgen (C4-2) in response to PARPi are elevated as a function of PCa progression, suggesting that the transcriptional regulatory functions of PARP-1 have clinical relevance.

5. The authors identify that HR gene expression alters in the course of disease progression based on TCGA data. They also identify mutations in HR genes. These data are descriptive and do not tell about the HR status during PCa development. Indeed, these authors have demonstrated that HR status is highly important to response to PARP inhibitors. Here, the authors should query how the HR function is affected during disease progression.

Several publications have previously demonstrated that the frequency of HR gene mutations increases as a function of prostate cancer progression (Grasso *et al.*, *Nature* 2012; Robinson *et al.*, *Cell* 2015; Gundem *et al.*, *Nature* 2015; Pritchard *et al.*, *Nature Commun* 2014; Pritchard *et al.*, *NEJM* 2016). However, none of these high-impact studies examined actual HR competency in these clinical specimens. Examining HR competency in clinical specimens representative of disease progression is beyond the scope of the current study.

The critical conclusion based on observations herein is that not only do HR gene defects increase during prostate cancer progression, the most frequent of these defects is mRNA upregulation. Since the data presented herein demonstrate that HR gene expression is controlled by PARP-1, and that PARP-1 enzymatic activity is increased during prostate cancer progression, there is an association

between PARP-1 activity and HR gene expression. We have modified the text on page 12 to better reflect our interpretation of the data.

Minor comment:

Abbreviation DHT should be explained first time it is used - dihydrotestosterone (DHT)

We have adjusted the text accordingly, and thank the reviewer for the editing suggestion.

Referee #3:

We are grateful for the suggestions posed by reviewer 3 and have addressed each comment in full, as described below:

1. While representing a remarkable body of work, and understanding the difficulty in working with PDX models and their inherent heterogeneity, the results of Fig 3 D and E are somewhat underwhelming. Please clarify whether the changes in D are statistically significant. Also, I'm not convinced that the 'majority' of changes in E are in the 'right' direction. Perhaps the author can elaborate on how they reached that conclusion.

We have clarified the text on page 9, having removed the word majority. Furthermore, we have conducted statistical analyses of these data (now included in this panel), as suggested by an expert biostatistician (Dr. Leiby). We performed the Wilcoxon signed rank test, and the results indicate the following with respect to the E2F1 target genes in Figure 3E: E2F1 ($p < 0.05$, down in 2/6 samples), PCNA ($p > 0.05$, down in 2/6 samples), MCM7 ($p < 0.01$, down in 4/6 samples), CCNA2 ($p < 0.01$, down in 4/6 samples). Overall, this statistical showed a significant difference between vehicle control and PARPi ($p < 0.0001$, 12 data points down-regulated, out of 24 total). As such, we are confident that these changes favor down-regulation in these patient tissues. While not requested, we performed similar statistical analyses on the HR expression data in the human explants depicted in revised Figure 5D, which demonstrates the PARPi elicits statistically significant down-regulation of HR genes in human prostate cancer tissue explants.

2. In Fig. 4.B, left, the overlap in the down-regulated genes appears highly significant. Thus, it would be important to clarify whether the enrichment shown in Fig 4.B, right, is based on standard GSEA,

or it's based on the hyper-enrichment test (i.e., the one available through GSEA here:

<https://na01.safelinks.protection.outlook.com/?url=http%3A%2F%2Fsoftware.broadinstitute.org%2Fgsea%2Fmsigdb%2Fannotate.jsp&data=02%7C01%7Ckaren.knudsen%40jefferson.edu%7C034040f37a7348ed1d4508d57f513d9d%7C55a89906c710436bbc444c590cb67c4a%7C0%7C0%7C636554909271630341&sdata=ySCxlpny8H4gosZNvqSyHuzGujGUCzDLsLTXZDHPX3I%3D&reserved=0>). I suspect it is the latter, since the authors include the title "Veliparib down-regulated only." If the former, however, it would be necessary, and reassuring, to show that GSEA on the Palbociclib down signature didn't return HR, etc. among the enriched genesets.

As described above in response to Reviewer 1, comment 13, we have added GSEA data for each subgroup utilizing the hyper-enrichment test. We have also performed statistical analyses of these data as described above (Chi-squared), which indicated these data sets are significantly different. As shown above, these data indicate that HR is uniquely enriched in the veliparib-responsive genes, confirming the original conclusion.

In Fig. 6B, if those reported are nominal p-values, it's inappropriate, and FDR q-values should be reported instead. Clearly, some of the q-values will not be significant. The authors can still make the case for an "HR enrichment" by testing the HR genes as a set by GSEA.

We have adjusted the figure to include FDR q-values.

Thank you for the submission of your revised manuscript to EMBO Molecular Medicine. We have now received the enclosed reports from the referees that were asked to re-assess it. As you will see the reviewers are now globally supportive and I am pleased to inform you that we will be able to accept your manuscript pending minor editorial amendments and a response to Referees #1 and #3.

***** Reviewer's comments *****

Referee #1 (Remarks for Author):

The authors have addressed the vast majority of my concerns; this is a really good paper! two minor issues remain:

1. In revised Figure S4B, gene expression of E2F1, PCNA, MCM7 and CCNA2 are shown, however only for the AdGFP (as the title of the bar graph indicates) and not for the E2F1 overexpression (AdE2F1). Please include these data.
2. Can the authors show the knockdown of E2F1 in revised Figure S6A?

Referee #2 (Remarks for Author):

The authors have improved the manuscript significantly and it is now a suitable study to publish

Referee #3 (Remarks for Author):

I believe the authors satisfactorily addressed the reviewers' comments and criticism, w/ a minor aspect still needing to be addressed.

Namely, the authors make a strong statement about the enrichment of the HR geneset in the "Valiparib unique" signature (Fig 4B), even including an arrow to highlight it, but they refrain from explicitly reporting the enrichment q-value, which appears not to be significant based on the color-coding. The authors should be explicit and report it, and if it's not (even marginally) significant, perhaps tone down the corresponding language.

2nd Revision - authors' response

10 October 2018

Referee #1:

1. In revised Figure S4B, gene expression of E2F1, PCNA, MCM7 and CCNA2 are shown, however only for the AdGFP (as the title of the bar graph indicates) and not for the E2F1 overexpression (AdE2F1). Please include these data.

We apologize for the error in labeling, but the data that were depicted in revised Figure S4B were actually derived from E2F1 overexpression. We have addressed the labeling, and included data from the control transduction (AdGFP), and we thank the reviewer for allowing us to correct this mistake.

2. Can the authors show the knockdown of E2F1 in revised Figure S6A?

We thank the reviewer for requesting that we depict this important control. We have now included the immunoblot demonstrating E2F1 knockdown in re-revised Figure S6A.

Referee #3:

1. The authors make a strong statement about the enrichment of the HR geneset in the "Valiparib unique" signature (Fig 4B), even including an arrow to highlight it, but they refrain from explicitly reporting the enrichment q-value, which appears not to be significant based on the color-coding. The authors should be explicit and report it, and if it's not (even marginally) significant, perhaps tone down the corresponding language.

The enrichment q value for the HR pathway in question is 0.0367, which met the pre-determined cut-off for significance. We agree that the color-coding made it difficult to determine whether these data were statistically significant. As such, we have opted to include the q values for each pathway and data set, as now depicted in re-revised Figure 4B. We thank the reviewer for the suggestion to increase the clarity of the data.

Corresponding Author Name: Karen E. Knudsen PhD

Manuscript Number: EMM-2017-08816-V2